# Model-Guided Dual-Role Alignment for High-Fidelity Open-Domain Video-to-Audio Generation

**Kang Zhang**[*1], **Trung X. Pham**[*1], **Suyeon Lee**[1], **Axi Niu**[2],
**Arda Senocak**[3], **Joon Son Chung**[1]
[1]KAIST, South Korea [2]NWPU, China [3]UNIST, South Korea
(zhangkang,trungpx,syl4356,joonson)@kaist.ac.kr
nax@nwpu.edu.cn, ardasnck@unist.ac.kr

## Abstract

We present MGAudio, a novel flow-based framework for open-domain video-to-audio generation, which introduces model-guided dual-role alignment as a central design principle. Unlike prior approaches that rely on classifier-based or classifier-free guidance, MGAudio enables the generative model to guide itself through a dedicated training objective designed for video-conditioned audio generation. The framework integrates three main components: (1) a scalable flow-based Transformer model, (2) a dual-role alignment mechanism where the audio-visual encoder serves both as a conditioning module and as a feature aligner to improve generation quality, and (3) a model-guided objective that enhances cross-modal coherence and audio realism. MGAudio achieves state-of-the-art performance on VGGSound, reducing FAD to 0.40, substantially surpassing the best classifier-free guidance baselines, and consistently outperforms existing methods across FD, IS, and alignment metrics. It also generalizes well to the challenging UnAV-100 benchmark. These results highlight model-guided dual-role alignment as a powerful and scalable paradigm for conditional video-to-audio generation. Code is available at:
https://github.com/pantheon5100/mgaudio

## 1 Introduction

Vision-guided audio generation is gaining increasing attention due to its critical role in Foley sound synthesis for video and film production [2]. In particular, video-to-audio (V2A) generation has emerged as a key task, not only for enriching silent videos produced by emerging text-to-video models [3, 4, 5, 6], but also for enhancing realism and immersion in professional video editing workflows. Realistic audio is essential for audio-visual coherence and immersive experience, but generating sound semantically aligned and temporally synchronized with video remains challenge.

Prior works have explored GAN-based [7] and autoregressive Transformer-based models [8], which often struggle with synchronization and semantic consistency. More recent methods adopt diffusion models to improve generation quality. For example, Diff-Foley [9] uses a contrastively trained video encoder to extract visual conditions, which

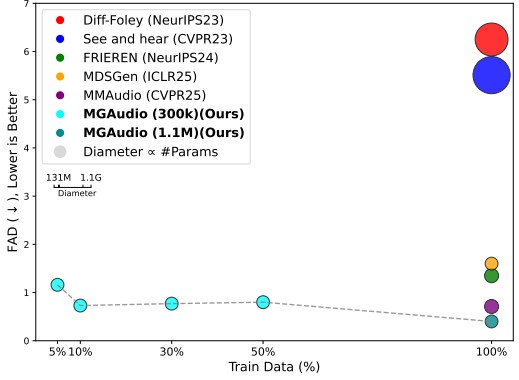

Figure 1: **V2A on VGGSound [1].** MGAudio attains the best FAD among video-to-audio methods with full data and 1.1M iters, and remains competitive even with only 10% data and 300k iters, highlighting its strong data efficiency.

---

*Equal contribution

39th Conference on Neural Information Processing Systems (NeurIPS 2025).

are then used to guide a diffusion model for audio generation. Other approaches, such as See and Hear [10] and FoleyCrafter [11], leverage large pretrained models to achieve high-fidelity results, but at the cost of massive model size (billions of parameters). In contrast, models like MDSGen [12], FRIEREN [13], and MMAudio [14] (131–159M parameters) explore more lightweight denoising or flow-matching [15] objectives, achieving competitive performance across fidelity and alignment metrics. Despite architectural differences, diffusion- and flow-matching–based methods typically rely on classifier-free guidance (CFG) to improve generation quality. This involves randomly dropping conditioning signals during training to simulate both conditional and unconditional objectives. While effective, this multi-task setup may dilute the model's capacity and lead to mismatched sampling behavior at inference.

To address the limitations of classifier-free guidance (CFG) in audio generation, we propose `MGAudio`, a novel *model-guided framework* that replaces CFG with direct model-based supervision during training. Our approach builds on the Model-Guided (MG) framework [16], originally introduced for class-conditional image generation on ImageNet, and effectively extends and adapts it to the video-to-audio domain. While MG is specified for discrete class labels, its application to conditional generation from continuous video inputs remains unexplored. To this end, `MGAudio` introduces a dual-role audio-visual encoder: one branch modulates the diffusion process via LayerNorm-based conditioning, while the other provides intermediate alignment signals to guide audio synthesis at multiple denoising steps.

We show that `MGAudio` (131M) achieves state-of-the-art performance on VGGSound [1], with a FAD of 0.40. Notably, the same model generalizes robustly to the UnAV-100 benchmark [17] without any fine-tuning, demonstrating strong cross-dataset transferability. Furthermore, when trained on just **10%** of the VGGSound, it outperforms existing methods trained on the full dataset across key metrics such as FAD (Fig. 1). These results demonstrate that model-guided training not only enables more efficient learning but also offers a compelling alternative to CFG for scalable and effective audio generation. Our key contributions are:

- We introduce `MGAudio`, a novel framework for video-to-audio generation that replaces the conventional *classifier-free guidance (CFG)* objective with a *model-guidance (MG)* objective, leading to more efficient training and improved generation quality.
- We propose a *dual-role alignment mechanism* that uses a shared jointly trained visual and audio encoder for conditioning and intermediate representation alignment, enabling more effective video-audio feature integration.
- `MGAudio` sets a new state-of-the-art on VGGSound, achieving an FAD of 0.4 with 131M parameters, and maintains strong performance with only 10% of the training data (FAD = 0.8), highlighting its efficiency and scalability.
- Extensive experiments and analyses on VGGSound and UnAV-100 benchmarks demonstrate that the model-guidance objective, combined with dual-role alignment, significantly improves data efficiency and generalization over prior methods.

## 2 Related Works

### 2.1 Classifier-Free Guidance Learning for Video-Guided Sound Generation

Early works such as SpecVQGAN [8] and Im2Wav [18] employ autoregressive Transformers to generate audio tokens conditioned on video or CLIP features. However, their sequential decoding results in slow inference and weak cross-modal alignment, limiting their effectiveness in open-domain scenarios. Recent advances in generative modeling have led to powerful diffusion-based methods that show strong performance across diverse application domains [19, 20]. Diff-Foley [9] introduces a two-stage pipeline combining contrastive video-audio pretraining with latent diffusion for more efficient generation. This has inspired follow-up work [10, 11]: [10] focuses on leveraging pre-trained generative models rather than training from scratch, while [11] targets improvements in audio-visual synchronization.

Despite their success, these approaches rely heavily on large U-Net backbones, which limits scalability and efficiency. To address this, MDSGen [12] incorporates recent innovations in generative modeling, such as Diffusion Transformers (DiT) [21] and spatial masking [22, 23], adapting masked diffusion Transformers for video-guided audio generation and achieving improved performance and efficiency.

The introduction of flow matching [15] has further expanded generative modeling abilities. Recent methods like FRIEREN [13] and MMAudio [14] adopt flow matching as an alternative to diffusion, using Transformer-based frameworks and reporting strong results on video-to-audio (V2A) tasks. Similarly, we adopt flow matching with a Transformer-based model.

However, despite architectural differences, most state-of-the-art approaches, whether diffusion, masked diffusion, or flow-based, continue to rely on *classifier-free guidance (CFG)* [24], where input conditions are randomly dropped during training (typically at a rate of $\eta = 10\%$) to enable multi-task learning (unconditional and conditional). In contrast, we shift the focus from this paradigm by adopting the recent Model Guided approach [16], moving toward a more efficient *audio model-guided (AMG)* strategy that achieves stronger generation quality and faster convergence.

## 2.2 Model-Guidance Learning in Audio Generation

Model-guided learning has recently been introduced in the vision domain as *Vision Model-Guidance* (VMG) [16] which enables efficient single-pass inference and is complementary to CFG, offering improvements in both inference modes. Inspired by this, we propose `MGAudio`, the *Audio Model-Guidance* (AMG) to audio generation, tailored for video-conditioned open-domain sound synthesis. Unlike the vision setting, we find that model-guided training alone improves convergence, but CFG remains beneficial at inference to ensure high-fidelity audio. This reveals a modality-specific behavior, suggesting that audio's temporal sensitivity may demand additional supervision at test time.

## 2.3 Feature Alignment for Accelerating Diffusion Learning

A key advancement in improving the training efficiency of vision diffusion is REPA [25], which proposes aligning intermediate image features in generative model with a pre-trained representation model [26, 27, 28, 29] to accelerate learning. Building on this idea, VA-VAE [30] and REPA-E [31] jointly optimize vision feature alignment in the VAE and the transformer, yielding notable gains in image generation quality. Inspired by these simple yet effective strategies, we introduce a novel approach for audio generation that aligns intermediate audio features from a strong pretrained audio model with the transformer model. We term this *dual-role audio-visual learning*, enabling efficient and semantically consistent generation through cross-modal feature supervision.

## 2.4 Temporal Alignment in Video-to-Audio Generation Methods

Another important research direction in video-to-audio generation focuses on enhancing temporal alignment between visual events and the generated audio. Recent works have explored various strategies to improve synchronization and cross-modal correspondence. AV-Link [32] employs temporally aligned self-attention fusion for bidirectional conditioning between audio and video. Hidden Alignment [33] enhances synchronization through encoder design and temporally aware augmentations. V2A-Mapper [34] adopts a lightweight approach that maps CLIP embeddings to CLAP space for modality bridging using foundation models.

More recent frameworks such as MDSGen [12], FRIEREN [13], and MMAudio [14] explore different ways of integrating video conditions: MDSGen [12] aggregates all frames into a compact global representation, FRIEREN [13] interpolates frame features along the temporal axis for alignment with the mel-spectrogram, and MMAudio [14] leverages a multimodal transformer to achieve fine-grained audio–video attention.

Although our work does not primarily focus on temporal conditioning design, we adopt the MDSGen-style global aggregation for simplicity and efficiency. As shown in Appendix J, different conditioning strategies each exhibit unique strengths across distributional, semantic, and temporal dimensions.

## 3 Method

We introduce `MGAudio`, the first audio generation framework to adopt model-guided training. Unlike classifier-free guidance (CFG) [24], which learns conditional model and unconditional model separately, model-guided learning provides direct supervision through the model itself, refining the model prediction directly aligns with CFG. As illustrated in Fig. 2, `MGAudio` comprises: (1) a *Dual-Role Audio-Visual Encoder* (DRAVE) with learnable projections for cross-modal alignment; (2) a *Flow-Based Denoising Transformer* (FBDT) that maps noise to audio via flow matching; and (3) an *Audio Model-Guidance* (AMG) training objective that enables replacing CFG with a more targeted,

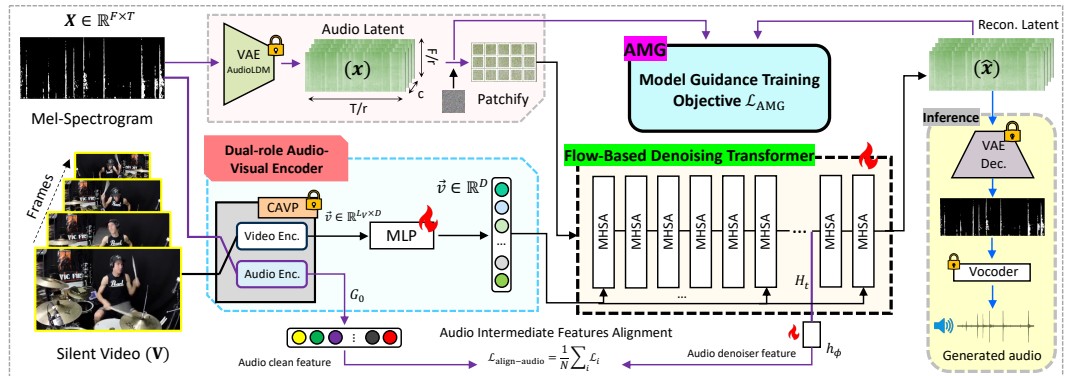

Figure 2: Overview of the `MGAudio` framework for video-guided audio generation. We design the first use of a model-guidance method that learns audio latent with a new objective and performs dual alignment learning. The violet arrow $\rightarrow$ is *training-only*, the black arrow $\rightarrow$ is used for both *training and inference*, and the blue arrow $\downarrow$ is *inference-only*. **MHSA**: *Multi-Head Self-Attention*.

model-driven signal. These components together enable robust alignment, efficient denoising, and high-quality video-conditioned audio generation. We describe each module below.

## 3.1 Scalable Flow-Based Denoising Transformer

*The first key component* of MGAudio is the *Flow-Based Denoising Transformer* (FBDT) shown in Fig. 2, which builds upon the scalable flow-matching architecture of the Scalable Interpolant Transformer (SiT) [35], a state-of-the-art generative modeling framework. In contrast to conventional diffusion models [36], flow matching learns continuous transport directions, yielding more stable and efficient denoising.

Given an input audio signal $\mathbf{A} \in \mathbb{R}^{L_\mathbf{A}}$ and a sequence of silent video frames $\mathbf{V} \in \mathbb{R}^{L_V \times 3 \times 224 \times 224}$ of length $L_V$, we first convert the $\sim 8s$ waveform into a mel-spectrogram $\mathbf{X} \in \mathbb{R}^{64 \times 816}$ using log-mel filterbanks with number of Mel bands 64. A pretrained AudioLDM VAE [37] $\mathcal{E}_{\text{VAE}}$ maps $\mathbf{X}$ into a latent representation $\mathbf{x} \in \mathbb{R}^{8 \times 16 \times 204}$, which is then patchified (patch size $p = 2$) and flattened to a sequence of tokens $\mathbf{x}' \in \mathbb{R}^{816 \times D}$ with $D = 768$ for the Base-size model. *In parallel*, silent video frames are processed by a video encoder $\mathcal{E}_{\text{Video}}$ (e.g., CAVP video encoder [9]) to produce frame-level embeddings $\mathbf{v} \in \mathbb{R}^{L_V \times 512}$. These features are temporally aggregated (e.g., via average pooling or attention, or with learnable $1 \times 1$ convolution layers as did in MDSGen [12]) to obtain a global video representation $\vec{v} \in \mathbb{R}^{1 \times D}$. This conditioning vector $\vec{v}$ guides the Transformer model $\theta$ to predict the noise vector $\epsilon$ or its equivalent flow direction $u$ during the reverse generation process. We formulate audio generation as a flow-matching problem [15], where the model learns the direction of transport from noise to data using a family of linear interpolants parameterized by time $t \in [0, 1]$. Following the flow-matching paradigm [15], we define a noisy latent at time $t$ as:

$$\mathbf{x}_t = (1 - t)\mathbf{x}_0 + t\boldsymbol{\epsilon}, \quad \boldsymbol{\epsilon} \sim \mathcal{N}(0, \mathbf{I}), \tag{1}$$

where $\mathbf{x}_0$ is the clean latent and $\boldsymbol{\epsilon}$ is standard Gaussian noise. The model is trained to predict the conditional flow direction that transports $\mathbf{x}_t$ toward $\mathbf{x}_0$:

$$\mathcal{L}_{\mathbf{FM}} = \mathbb{E}_{t,\mathbf{x}_0,\vec{v},\boldsymbol{\epsilon}} \|u_\theta(\mathbf{x}_t, \vec{v}, t) - u_t(\mathbf{x}_t | \mathbf{x}_0)\|^2, \tag{2}$$

where $u_\theta$ is the model-predicted flow direction. The true flow $u_t$ is analytically derived from the derivative of the interpolant $u_t(\mathbf{x}_t | \mathbf{x}_0) = \mathbf{x}_0 - \boldsymbol{\epsilon}$. This flow-matching loss serves as the core denoising objective. Its efficacy is further enhanced by our model-guided formulation (Sec. 3.3), which refines the supervision signal to better align with video context.

During inference, the model starts from pure Gaussian noise $\mathbf{x}_1 \sim \mathcal{N}(0, I)$ in the latent space and iteratively denoises using the predicted flows, progressively refining the latent $\hat{\mathbf{x}} \in \mathbb{R}^{8 \times 16 \times 204}$. The final mel-spectrogram $\widehat{\mathbf{X}} \in \mathbb{R}^{64 \times 816}$ is recovered using the VAE decoder, and a neural vocoder [38] reconstructs the corresponding waveform. This formulation combines the scalability and efficiency of flow-based models with the expressive power of Transformer architectures and robust conditioning mechanisms for open-domain video-to-audio generation.

## 3.2 Dual-Role Audio-Visual Encoder

*The second key component* of our framework is the Dual-Role Audio-Visual Encoder (DRAVE), illustrated in Fig. 2, which comprises two main branches: (1) an audio encoder and (2) a video encoder. We adopt the Contrastive Audio-Visual Pretraining (CAVP) strategy introduced in Diff-Foley [9], which effectively aligns audio and video representations and has demonstrated strong performance in models like FRIEREN [13] and MDSGen [12]. *While prior works typically use only the CAVP video encoder, discarding the audio counterpart, we leverage both.* Specifically, we use the CAVP video encoder to condition the denoising process for *vision-audio alignment* (*Video Processor* branch) and simultaneously integrate the CAVP audio encoder for *audio-audio alignment* (*Audio Processor* branch), drawing inspiration from REPA [25] in the vision domain. This dual-role design enhances representation learning and significantly improves audio generation quality.

**Video Processor.** As illustrated in Fig. 2, the input condition for the V2A task is a silent video. We extract frame-level features using the *CAVP video encoder* [9], resulting in embeddings $\mathbf{v} \in \mathbb{R}^{L_V \times 512}$. These are projected to $\mathbb{R}^{L_V \times 768}$ via a multi-layer perceptron (MLP) to match the input dimension of the Transformer's multi-head self-attention layers (model size B). To reduce temporal redundancy and aggregate contextual cues, we follow MDSGen [12] and apply a $1 \times 1$ convolution to condense the sequence into a compact global feature vector $\vec{v} \in \mathbb{R}^{1 \times 768}$. This vector serves as a conditioning signal in the denoising process, injected through Adaptive LayerNorm (AdaLN) to guide the generation process with visual context.

**Audio Processor.** The ground-truth mel-spectrogram $\mathbf{X}$ is processed via two parallel branches, as shown in Fig. 2. In the first branch, $\mathbf{X}$ is encoded by the AudioLDM2 VAE [38], where Gaussian noise is injected into the latent space to facilitate denoising-based generation. In the second branch, the same spectrogram is passed through the *CAVP audio encoder* [9] to provide an auxiliary representation for regularizing the learning process.

Drawing inspiration from regularization techniques in the vision domain, such as REPA [25] and VA-VAE [30], we adopt a similar strategy for cross-modal alignment in audio generation. Unlike REPA, which relies on generic visual encoders like DINOv2, we find that CAVP, specifically trained for audio-visual correspondence, significantly improves alignment and synthesis quality. Our ablation studies confirm that the CAVP audio-visual encoder plays a dual role: facilitating alignment with video features and enhancing intermediate representations when paired with the denoising transformer. We define the alignment loss used for joint training with the denoising branch as:

$$\mathcal{L}_{\text{align-audio}} = -\mathbb{E}_{\mathbf{x},\epsilon,t} \left[ \frac{1}{\mathcal{B}} \sum_{i=1}^{\mathcal{B}} \mathcal{L}_i \right], \quad \text{with } \mathcal{L}_i = \text{similarity}\big(\mathbf{G}_0^i, \ h_\phi(\mathbf{H}_t^i)\big), \tag{3}$$

where $\mathcal{L}_i$ denotes the alignment loss for the $i$-th audio patch among $\mathcal{B}$ total patches. $\mathbf{G}_0$ represents the CAVP-encoded clean audio features, and $\mathbf{H}_t$ is the intermediate latent from the noisy audio branch at time step $t$. $\mathbf{H}_t$ is projected through an MLP $h_\phi$, to match the $\mathbf{G}_0$'s dimensionality. Following REPA [25], we use cosine similarity and retain the same number of positional layers for consistency. *We avoid applying an additional MLP to $\mathbf{G}_0$, as it led to feature collapse in our experiments.*

**Overall Training Objective.** The overall training loss combines the flow-matching denoising loss and the alignment loss, balanced by a scaling factor $\lambda$, which we set to 0.5 by default, following prior work [25]. The total objective is defined as:

$$\mathcal{L}_{\text{FM-align}} = \mathcal{L}_{\mathbf{FM}} + \lambda \mathcal{L}_{\text{align-audio}}. \tag{4}$$

Here, $\mathcal{L}_{\mathbf{FM}}$ denotes the flow-matching loss for denoising, while $\mathcal{L}_{\text{align-audio}}$ enforces consistency between pretrained clean audio and intermediate noisy audio generator representations.

## 3.3 Audio Model-Guidance (AMG)

*The third core component* of MGAudio is the *Audio Model-Guidance* (AMG) mechanism, which departs from traditional *classifier-free guidance* (CFG) approaches commonly used in audio generation. As illustrated in Fig. 2, AMG serves as the key to effectively training our model under video-conditioned audio generation. Building on the FBDT backbone, AMG replaces CFG with a self-distilled target. Let $u_\theta(\mathbf{x}_t, \vec{v}, t)$ be the conditioned flow prediction and $u_\theta(\mathbf{x}_t, \varnothing, t)$ be the

unconditioned one. Our AMG objective is defined as a variant of VMG [16] that reformulates the training target to directly approximate the CFG optimization trajectory:

$$\mathcal{L}_{\mathbf{AMG}} = \mathbb{E}_{t,\mathbf{x}_0,\vec{v},\boldsymbol{\epsilon}} \|u_\theta(\mathbf{x}_t, \vec{v}, t) - u'\|^2, \tag{5}$$

where the model-guided target $u'$ is defined as:

$$u' = u + w \cdot \text{sg}\left(u_\theta(\mathbf{x}_t, \vec{v}, t)\right) - u_\theta(\mathbf{x}_t, \varnothing, t), \tag{6}$$

where $w$ denotes the guidance scale factor, and $\text{sg}(\cdot)$ is the stop-gradient operator, used to block gradient flow through the guidance term. This stabilizes training and prevents degenerate solutions, as shown in [16]. To further improve stability, we compute $u'$ using an Exponential Moving Average (EMA) version of the online model $u_\theta$, ensuring more reliable target predictions over the course of training. During training, the condition vector $\vec{v}$ is randomly replaced with a zero vector $\varnothing$ with probability $\psi$. The final loss in Eq. 4 training objective for MGAudio becomes:

$$\mathcal{L}_{\text{FM-align}} = \mathcal{L}_{\mathbf{AMG}} + \lambda \mathcal{L}_{\text{align-audio}}. \tag{7}$$

### 3.4 AMG Meets CFG: Toward Efficient High-Fidelity Audio Generation

*Finally*, while model-guided training alone has proven effective in the vision domain, e.g., VMG [16] achieves state-of-the-art performance without external guidance and sees only marginal gains when combined with CFG, we observe a distinct trend in audio generation. Specifically, although training with *Audio Model-Guidance* (AMG) already surpasses baseline models without guidance (e.g., SiT), the highest audio generation quality is achieved when AMG-trained models are further enhanced with classifier-free guidance (CFG) at inference.

We provide comprehensive ablations and analyses of this synergy in the experiments section. Notably, we find that conventional CFG training, i.e., randomly dropping a fraction of conditioning inputs during training as defined in Eq. 2, performs worse than our AMG formulation in Eq. 5, which optimizes an implicit classifier via a modified objective. This demonstrates the superiority of AMG as a training strategy, not just as a guidance method at inference time.

*Lastly*, motivated by the robustness and data efficiency observed in AMG-based training, we perform a low-resource experiment on VGGSound. Remarkably, MGAudio achieves strong performance across multiple evaluation metrics using only **10%** of the full training set. We explore this strong generalization capability in the ablation section, where we hypothesize that AMG promotes highly efficient learning by effectively leveraging the task's conditional structure. Our analysis also reveals that *data quality, rather than quantity*, remains a critical bottleneck in VGGSound.

## 4 Experiments

### 4.1 Implementation Details

**Datasets and Metrics.** We train on the VGGSound dataset [1], which contains in-the-wild video clips from YouTube, with ∼182k for training and ∼15k for testing. For generalization, we also test on the UnAV-100 dataset [17], which includes 10,791 test videos with annotated sound events. Following prior works [14, 13, 12], we evaluate generation quality using Fréchet Distance (FD), Fréchet Audio Distance (FAD), Inception Score (IS), KL divergence, and audio-video alignment accuracy.

**Training.** MGAudio is trained for 1.1M steps with a batch size of 64, learning rate of 1e-4, and guidance scale $w = 1.45$. All experiments are run on a single A100 (80GB). For all experiment we use sampling step of 50 and CFG value of 1.45. Following [13, 12], video-audio pairs are truncated to 8.2 seconds. We use Base model size for its performance-efficiency tradeoff. Additional details of experimental setup is in Appendix.

### 4.2 Main Results

**a) VGGSound Dataset.** Tab. 1 shows that MGAudio, the first model-guided (MG) audio generation framework, achieves state-of-the-art results across multiple metrics using only 131M parameters. It obtains an FAD of 0.40, significantly outperforming the second-best, MMAudio [14]. With a CFG scale of 4.0, MGAudio achieves 99.04% alignment accuracy, surpassing all baselines. This metric, introduced by [9], evaluates temporal synchronization and semantic coherence using a dedicated

Table 1: `MGAudio` **on VGGSound.** Audio generation quality across state-of-the-art methods on the VGGSound test set is presented. Bold indicates the best performance, and an underline denotes the second-best. CFG: *Classifier-Free Guidance*, MG: *Model-Guidance*. *We train MMAudio from scratch solely on the VGGSound dataset, and set the text input to *None* during inference for fairness.

| Method | FAD↓ | FD↓ | IS↑ | KL↓ | Align. Acc.↑ | Time↓ (s) | # Params↓ | Train Type |
|---|---|---|---|---|---|---|---|---|
| Diff-Foley [NeurIPS'23] [9] | 6.25 | 23.07 | 10.85 | 3.18 | 93.94 | 0.36 | 860M | CFG |
| See and Hear [CVPR'24] [10] | 5.51 | 26.60 | 5.47 | 2.81 | 58.14 | 18.25 | 1099M | CFG |
| FRIEREN [NeurIPS'24] [13] | 1.38 | 12.36 | 12.12 | 2.73 | 97.25 | **0.20** | 157M | CFG |
| MDSGen [ICLR'25] [12] | 1.40 | 17.42 | 9.66 | 2.84 | 96.88 | 0.24 | **131M** | CFG |
| MMAudio [CVPR'25]* [14] | 0.71 | 6.97 | 11.09 | **2.07** | 92.28 | 0.98 | 157M | CFG |
| **MGAudio**, CFG = 1.45 | **0.40** | **6.16** | 12.82 | 2.76 | 95.65 | 0.31 | **131M** | MG |
| **MGAudio**, CFG = 4.0 | 2.23 | 13.65 | 12.68 | 2.72 | **99.04** | 0.31 | **131M** | MG |
| **MGAudio**, CFG = 6.0 | 1.50 | 9.10 | **17.39** | 2.80 | 98.10 | 0.31 | **131M** | MG |

classifier. At a higher CFG scale (6.0), MGAudio also reaches an IS of 17.39, though with some trade-offs in other metrics. Overall, these results highlight the efficiency of model-guided training over conventional classifier-free guidance.

Table 2: `MGAudio` **on UnAV-100.** All models are trained on VGGSound from Tab. 1 and evaluated on UnAV-100 test set for generalization. Bold indicates the best, and an underline for the second-best.

| Method | FAD↓ | FD↓ | IS↑ | KL↓ | Align. Acc.↑ | Train Type |
|---|---|---|---|---|---|---|
| Diff-Foley [NeurIPS'23] [9] | 7.51 | 24.25 | 10.74 | 2.46 | 87.46 | CFG |
| MDSGen [ICLR'25] [12] | 2.01 | 16.10 | 10.10 | 2.18 | 97.57 | CFG |
| FRIEREN [NeurIPS'24] [13] | 1.40 | 10.82 | 13.46 | 2.02 | **98.14** | CFG |
| MMAudio [CVPR'25] [14] | 0.93 | 8.63 | 11.37 | **1.62** | 85.68 | CFG |
| **MGAudio (Ours)** | **0.54** | **5.40** | **13.90** | 2.00 | 97.54 | MG |

**b) UnAV-100 Dataset.** To assess generalization of `MGAudio`, we evaluate on UnAV-100 [17]. Tab. 2 presents the results. All models are models trained on VGGSound as reported in Tab. 1. `MGAudio` achieves the best performance across FAD, FD, and IS. While MMAudio gives the lowest KL divergence, it suffers from poor alignment accuracy. In contrast, `MGAudio` maintains strong alignment accuracy (97.54%). These results emphasize the robustness and strong generalization of our model-guided approach, even on datasets with dense, diverse audio-visual events not seen during training.

### 4.3 Ablation Study

We attribute `MGAudio`'s strong performance to three design choices: the Audio Model-Guidance (AMG) objective for improved supervision, a Dual-Role Audio-Visual Encoder that fully exploits CAVP's alignment strength, and compatibility with CFG at inference despite being trained without it. We perform five ablations under a unified setup, including a data efficiency study showing AMG's robustness with just 10% of VGGSound. Comparisons with CFG-only baselines and various alignment encoders (DINOv2, CLAP, CAVP) highlight the benefits of dual-alignment and the trade-offs in fidelity. We also demonstrate scalability across model sizes and show via UMAP [39] that AMG produces tighter, more class-consistent audio distributions than CFG-based methods. All ablations use 300k steps and a smaller batch size of 16 for faster iteration.

#### 4.3.1 Efficiency of Data Utilization

Motivated by the strong generalization and data efficiency of AMG training (Tab. 1, 2), we conduct a low-resource experiment on VGGSound with only 300k training steps. Interestingly, as shown in Tab. 3, `MGAudio` achieves comparable or even superior performance using just **10%** of the training data, surpassing the full-data model on most metrics. Furthermore, compared with Table 1, our model trained with only 10% of the data already outperforms Diff-Foley, See and Hear, FRIEREN, and MDSGen in terms of FAD, FD, and KL, and achieves performance comparable to MMAudio. This highlights AMG's ability to effectively exploit conditional structure for efficient learning.

A manual inspection of 100 random samples in VGGSound reveals that approximately 6% exhibit weak or irrelevant audio-visual alignment, and 9% contain static frames or silence, leaving only 85% with meaningful, well-aligned content. This suggests that noisy supervision in VGGSound may hinder performance, and in some cases, less but cleaner data can lead to better results. Nonetheless, we use the full dataset for all main experiments to ensure fair comparisons with prior work.

Table 3: **Data Efficiency**. Interestingly, training `MGAudio` on just 10% of VGGSound yields strong performance, surpassing the full-data setting on several metrics. This highlights the model's data efficiency and robustness. AA: Alignment Accuracy (%).

| Train Data | VGGSound [1] | | | | | UnAV-100 [17] | | | | |
|---|---|---|---|---|---|---|---|---|---|---|
| | FAD↓ | FD↓ | IS↑ | KL↓ | AA ↑ | FAD↓ (s) | FD↓ | IS↑ | KL↓ | AA ↑ |
| 5% | 1.16 | 9.72 | 9.80 | 2.67 | 93.37 | 1.37 | 7.76 | 10.94 | 1.95 | 96.00 |
| **10%** | **0.73** | **8.79** | **10.28** | **2.64** | **95.73** | 0.85 | **7.15** | **11.67** | **1.95** | **97.61** |
| 30% | 0.77 | 10.17 | 9.83 | 2.69 | 95.27 | **0.73** | 8.29 | 11.09 | 2.04 | 97.42 |
| 50% | 0.80 | 10.44 | 9.35 | 2.69 | 94.51 | 0.78 | 8.39 | 10.50 | 2.03 | 96.97 |
| 100% | 0.81 | 10.25 | 9.85 | 2.71 | 95.14 | 0.89 | 9.88 | 9.72 | 2.09 | 96.60 |

### 4.3.2 Effect of Model-Guidance and Dual-Alignment

We compare our model-guided approach with the CFG-based baseline SiT [35] in Tab. 4. SiT achieves strong performance on V2A with CFG, notably an FAD of 1.52. However, while model guidance (MG) alone underperforms, combining MG with CFG at inference significantly boosts results, surpassing SiT across all metrics. This contrasts with findings in the ImageNet domain, where MG alone is sufficient. Finally, our proposed dual-alignment strategy further improves performance, outperforming SiT with CFG.

Table 4: **Effectiveness of Model Guidance.** Evaluation on the VGGSound test set shows that combining model-guidance (MG) with classifier-free guidance (CFG) at inference significantly boosts performance. Our dual-alignment strategy further enhances learning across all metrics.

| Setting | CFG | Model-Guide | Dual-Alignment | FAD↓ | FD↓ | IS↑ | KL↓ | AA |
|---|---|---|---|---|---|---|---|---|
| (a) No Guidance | - | - | - | 2.67 | 17.23 | 5.69 | 3.01 | 78.20 |
| (b) SiT [35] | ✓ | - | - | 1.52 | 16.25 | 6.51 | 2.87 | 87.07 |
| (c) MG [16] | - | ✓ | - | 2.13 | 16.20 | 6.02 | 2.85 | 83.87 |
| (d) Joint Guidance | ✓ | ✓ | - | 1.19 | 14.26 | 7.70 | 2.73 | 92.37 |
| (e) **MGAudio** | ✓ | ✓ | ✓ | **1.14** | **13.09** | **8.35** | **2.70** | **93.67** |

### 4.3.3 Choice of Alignment Encoders

We find that prior works often use the video encoder as the only conditioning signal, without fully leveraging its potential for joint integration in VAE or transformer alignment. In our analysis, we focus on the role of the audio encoder to demonstrate how deeper integration can improve performance. While REPA [25] leverages DINOv2 [26] for representation alignment in

Table 5: **Impact of Audio Encoder Choice.** Evaluation on the VGGSound test set shows that CAVP yields a better FAD score. Gray indicates default.

| Encoder | FAD↓ | FD↓ | IS↑ | KL↓ | Align. Acc. |
|---|---|---|---|---|---|
| (a) DINOv2 | 5.52 | 27.17 | 3.25 | 4.34 | 21.66 |
| (b) CLAP | 1.35 | **12.23** | 8.27 | **2.68** | 93.19 |
| (c) CAVP | **1.14** | 13.09 | **8.35** | 2.70 | **93.67** |

vision tasks, directly applying MG with REPA and DINOv2 yields subpar results in video-to-audio synthesis. Instead, we investigate various audio encoders and observe that CAVP [9] and CLAP [40] achieve the best performance. Notably, CAVP attains a lower FAD of 1.14 compared to CLAP's 1.35, with other metrics remaining comparable. For simplicity, we adopt CAVP, which is already available via Diff-Foley's public checkpoint, avoiding the need for additional encoder models. As shown in Fig. 3, using DINOv2 for alignment often leads to semantically incorrect generations, for instance, generating speech instead of impact sounds in a badminton video, or failing to synthesize audio for musical performances. In contrast, CAVP and CLAP produce more coherent and relevant results.

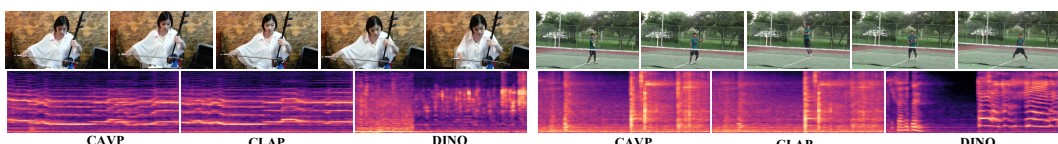

Figure 3: **Effect of Alignment Encoder in Mel-Spectrogram.** The selection of alignment encoders significantly impacts the quality of generated audio in the V2A task.

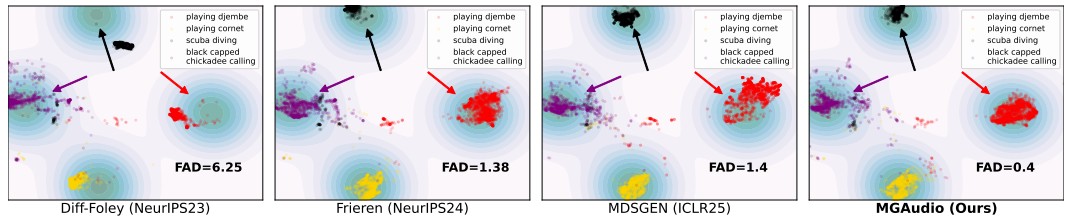

Figure 4: **Audio Distribution.** `MGAudio` generates audio samples that *more closely align* with the target distribution compared to other methods. For example, samples for classes "*playing djembe*" (red points) and "*scuba diving*" (black points) are tightly clustered around the center of the real sample distribution. Full-resolution version are provided in the supplementary for clearer inspection.

### 4.3.4 Scalability

We demonstrate that `MGAudio` scales in model size effectively across key metrics: FAD, FD, and IS, as shown in Tab. 6. We evaluate four model sizes: **S/2, B/2, L/2, XL/2**, where "/2" refers to the default patch size used in SiT [35]. Unlike MDSGen [12], which reported overfitting in their L2 model, our model-guided approach maintains strong performance even at larger scales. To balance performance and effi-

Table 6: **Scalability.** Evaluation on the VGGSound test set shows that `MGAudio` maintains strong performance as model size increases, demonstrating effective scalability with larger parameter counts.

| Model Size | FAD↓ | FD↓ | IS↑ | KL↓ | Align. Acc. |
|---|---|---|---|---|---|
| S/2 (34M) | 2.54 | 18.14 | 6.23 | 3.13 | 93.53 |
| B/2 (131M) | 1.14 | 13.09 | 8.35 | **2.70** | 93.67 |
| L/2 (464M) | 0.95 | 10.22 | 9.84 | 2.73 | **94.50** |
| XL/2 (680M) | **0.90** | **10.09** | **9.96** | 2.73 | 94.40 |

ciency, we adopt the **B/2-size model (131M)** in all main experiments, as it already delivers competitive results with significantly lower computational cost compared to baselines.

### 4.3.5 Audio Distribution Learned by Model Guidance

We compare the generative behavior of our model-guided `MGAudio` with classifier-free guidance (CFG)-based methods, including Diff-Foley [9], FRIEREN [13], and MDSGen [12]. To evaluate the learned distributions, we select four distinct VGGSound classes: *playing djembe, playing cornet, scuba diving, and black capped chickadee calling*. For each class, we sample 50 videos and generate 20 outputs per video (1000 samples per method) using varied seeds. The distributions are visualized via UMAP [39]. As shown in Fig. 4, `MGAudio` produces samples that cluster more tightly around the real sample distributions than other methods, reflecting stronger class consistency and diversity. These observations align with the quantitative results in Tab. 1, where `MGAudio` achieves a significantly lower FAD than the others. FD metrics further support these findings.

### 4.3.6 Model-Guidance vs. Classifier-Free Guidance

We examine the effect of classifier-free guidance (CFG) during inference for models trained with different objectives. Here, we use sampling step of 25 same with FRIEREN. Fig. 5 shows that FRIEREN [13], which depends on CFG during both training and inference, suffers significantly when CFG is disabled (i.e., CFG = 1). In contrast, `MGAudio`, trained with our AMG (model-guided) objective, performs strongly even without CFG. Applying an optimal CFG scale further improves `MGAudio`, setting a new state-of-the-art. These results highlight the robustness of model-guided training and its reduced reliance on inference-time CFG for effective video-to-audio generation.

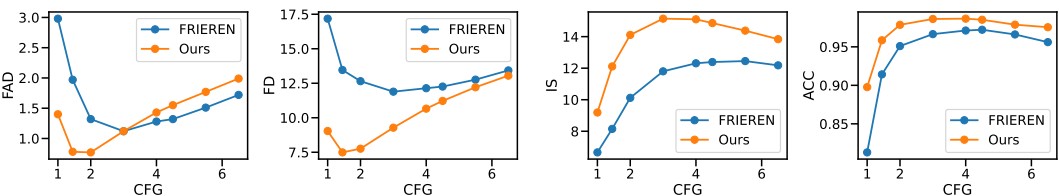

Figure 5: **Effect of CFG vs. AMG.** Training with AMG consistently outperforms CFG across all evaluation metrics on the video-to-audio task, highlighting the advantages of model-guided learning.

To further validate these findings under controlled conditions, we compare MGAudio with the CFG-only baseline SiT, which shares the same architecture, training setup, and optimization parameters. Both models are trained for 300,000 iterations with a batch size of 32. Figure 6 presents FAD and FD results across various sampling steps, with and without CFG applied at inference. MGAudio consistently outperforms the SiT baseline in all configurations. Notably, even without CFG—which reduces inference cost by eliminating the need for dual model passes—MGAudio achieves superior perceptual and distributional quality. This confirms that model-guided training not only enhances robustness and efficiency but also provides consistent gains across both comparable baselines (SiT) and prior state-of-the-art methods (FRIEREN).

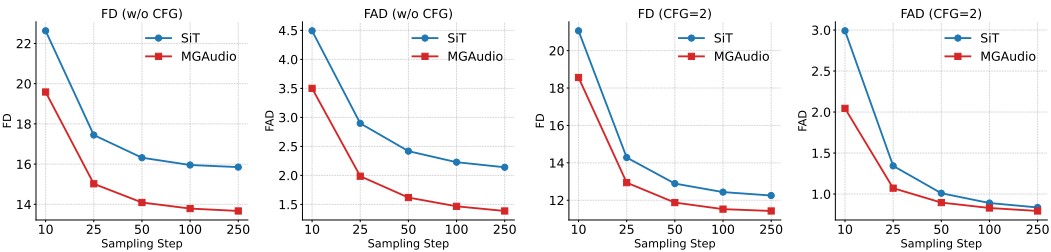

Figure 6: Comparison of FD and FAD metrics for SiT and MGAudio models, evaluated with and without CFG across different sampling steps.

## 4.4 Limitations

One limitation of our method is its reduced effectiveness when generating human vocalizations or linguistically complex audio, such as dialogue or singing, especially from subtle visual cues like lip movements. Since MGAudio is optimized for general audio effects and ambient sounds, it lacks the structural and phonetic awareness needed to model such content reliably. Additionally, while model-guided training improves overall alignment and diversity, it may introduce confusion when the visual semantics are ambiguous or loosely correlated with the target sound. Furthermore, inference speed remains a bottleneck due to the reliance on a VAE and the iterative sampling process. A promising direction for future work is to adopt acceleration techniques such as Consistency Models [41] or Physics-Informed Distillation (PID) [42].

## 5 Conclusion

In this work, we reveal a compelling property of the proposed MGAudio, a model based on scalable interpolant transformers (SiT) trained with model-guided objectives, demonstrating their strong potential for efficient audio generation. Our MGAudio framework achieves state-of-the-art performance on VGGSound with a FAD of 0.40, which sets a new benchmark. Notably, training with only 10% of the VGGSound dataset surpassing most existing methods that rely on 100% of the data in terms of FAD. We also show that model-guidance produces a more compact and coherent audio distribution compared to classifier-free guidance (CFG). Finally, by aligning audio and video representations through dual learning, our framework utilizes pretrained video encoders more effectively than conventional conditioning strategies. These contributions collectively highlight the potential of model-guided diffusion transformers as a scalable, data-efficient solution for multimodal audio generation.

## Acknowledgments and Disclosure of Funding

This work was supported by the IITP grant funded by the Korean government (MSIT, RS-2025-02215122, Development and Demonstration of Lightweight AI Model for Smart Homes, 90%) and partially supported by the KAIST Jang Young Sil Fellow Program (10%).

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

# Model-Guided Dual-Role Alignment for High-Fidelity Open-Domain Video-to-Audio Generation

# Appendix

## A Model Guidance: Derivation and Integration

In our framework, MGAudio, we incorporate the *Model Guidance* (MG) strategy proposed by [16] as a training-time objective that complements the conventional Classifier-Free Guidance (CFG). While CFG modifies the inference trajectory by combining outputs from conditional and unconditional branches, MG introduces an auxiliary loss that explicitly accounts for the posterior dependency between conditions and noisy inputs. This enables improved condition alignment without incurring additional inference-time overhead.

### A.1 Model Guidance for Flow Matching

Our model is based on the flow-matching mechanism [15] for the denoising process, main backbone being SiT [35]. Flow-Matching aims to learn a conditional velocity field $u_\theta(\mathbf{x}_t, t, \vec{v})$ given condition $\vec{v}$ (condense feature vector extracted from the silent video), such that it matches the ground-truth flow:

$$u_t(\mathbf{x}_t \mid \mathbf{x}_0) = \frac{\mathrm{d}\mathbf{x}_t}{\mathrm{d}t} = \mathbf{x}_0 - \boldsymbol{\epsilon}, \tag{8}$$

where $\boldsymbol{\epsilon} \sim \mathcal{N}(0, \mathbf{I})$ and $\mathbf{x}_t = (1 - t)\mathbf{x}_0 + t\boldsymbol{\epsilon}$. The standard flow-matching loss is:

$$\mathcal{L}_{\mathrm{FM}} = \mathbb{E}_{\mathbf{x}_0, \boldsymbol{\epsilon}, t, \vec{v}} \|u_\theta(\mathbf{x}_t, t, \vec{v}) - u_t(\mathbf{x}_t \mid \mathbf{x}_0)\|_2^2. \tag{9}$$

While Eq. 9 learns to model the conditional distribution $p_\theta(\mathbf{x}_t \mid \vec{v})$, diffusion models often underutilize the conditioning information in practice. To address this, MG proposes to include the posterior term $p_\theta(\vec{v} \mid \mathbf{x}_t)$, resulting in the joint distribution:

$$\tilde{p}_\theta(\mathbf{x}_t \mid \vec{v}) = p_\theta(\mathbf{x}_t \mid \vec{v}) \cdot p_\theta(\vec{v} \mid \mathbf{x}_t)^w, \tag{10}$$

where $w$ is the guidance scale controlling the strength of the posterior term. The score of this joint distribution becomes:

$$\nabla_{\mathbf{x}_t} \log \tilde{p}_\theta(\mathbf{x}_t \mid \vec{v}) = \nabla_{\mathbf{x}_t} \log p_\theta(\mathbf{x}_t \mid \vec{v}) + w \cdot \nabla_{\mathbf{x}_t} \log p_\theta(\vec{v} \mid \mathbf{x}_t). \tag{11}$$

Using Bayes' rule, we express the posterior as:

$$\log p_\theta(\vec{v} \mid \mathbf{x}_t) \propto \log p_\theta(\mathbf{x}_t \mid \vec{v}) - \log p_\theta(\mathbf{x}_t), \tag{12}$$

which leads to the posterior gradient:

$$\nabla_{\mathbf{x}_t} \log p_\theta(\vec{v} \mid \mathbf{x}_t) \propto u_\theta(\mathbf{x}_t, t, \emptyset) - u_\theta(\mathbf{x}_t, t, \vec{v}), \tag{13}$$

where $u_\theta(\mathbf{x}_t, t, \emptyset)$ denotes the velocity predicted without conditioning.

This gives rise to the modified target velocity:

$$\mathbf{u}' = \mathbf{u} + w \cdot \mathrm{sg}\left(u_\theta(\mathbf{x}_t, t, \vec{v}) - u_\theta(\mathbf{x}_t, t, \emptyset)\right), \tag{14}$$

where $\mathbf{u} = \mathbf{x}_0 - \boldsymbol{\epsilon}$ is the ground-truth velocity and $\mathrm{sg}(\cdot)$ denotes the stop-gradient operation, used to stabilize training. The final Model Guidance loss is:

$$\mathcal{L}_{\mathrm{MG}} = \mathbb{E}_{\mathbf{x}_0, \boldsymbol{\epsilon}, t, \vec{v}} \|u_\theta(\mathbf{x}_t, t, \vec{v}) - \mathbf{u}'\|_2^2. \tag{15}$$

Table 7: `MGAudio` **on VGGSound.** Audio generation quality across state-of-the-art methods on the VGGSound test set is presented. Bold indicates the best performance, and an underline denotes the second-best. CFG: *Classifier-Free Guidance*, MG: *Model-Guidance*. *We train MMAudio from scratch solely on the VGGSound dataset, and set the text input to *None* during inference for fairness.

| Method | Train Type | Inference Type | FAD↓ | FD↓ | IS↑ | KL↓ | Align. Acc.↑ |
|---|---|---|---|---|---|---|---|
| Diff-Foley [NeurIPS'23] [9] | CFG | CFG | 6.25 | 23.07 | 10.85 | 3.18 | 93.94 |
| See and Hear [CVPR'24] [10] | CFG | CFG | 5.51 | 26.60 | 5.47 | 2.81 | 58.14 |
| FRIEREN [NeurIPS'24] [13] | CFG | CFG | 1.38 | 12.36 | 12.12 | 2.73 | 97.25 |
| MDSGen [ICLR'25] [12] | CFG | CFG | 1.40 | 17.42 | 9.66 | 2.84 | 96.88 |
| MMAudio [CVPR'25]* [14] | CFG | CFG | 0.71 | 6.97 | 11.09 | **2.07** | 92.28 |
| **MGAudio** | **MG** | **No CFG** | 0.80 | 7.89 | 9.90 | 2.77 | 90.80 |
| **MGAudio** | **MG** | **CFG** | **0.40** | **6.16** | **12.82** | 2.76 | 95.65 |

## A.2 Inference-Time Integration of CFG into MG-Trained Models

At inference, we integrate classifier-free guidance (CFG) to explicitly control generation fidelity. Although the MG loss is utilized solely during training to enhance model sensitivity to conditioning signals, employing CFG at inference further boosts fidelity without additional retraining. Empirical results demonstrate that combining MG with CFG achieves superior perceptual quality and alignment, effectively balancing diversity and fidelity across key metrics such as FAD.

Tab. 7 shows that our best MGAudio model (131M parameters), trained for 1.1M steps without CFG, achieves a competitive FAD score of 0.80 and alignment accuracy of 90.80%, outperforming strong baselines like MDSGen (131M parameters) and FRIEREN (157M parameters), and closely approaching MMAudio (157M parameters), all utilizing CFG at inference. Notably, applying CFG significantly enhances our model's performance, achieving state-of-the-art results with an FAD of 0.40 and alignment accuracy of 95.65%, underscoring CFG's critical role in improving alignment quality.

## B  More Training Details

**Training Configurations.** We adopt the same model architecture and hyperparameters as SiT [35]. For all experiments, the guidance scale factor in Eq. 6 of the main paper is set to $w = 1.45$, following the default in [16]. For the initial 10,000 training steps, we set $w = 0$ to stabilize early training. All models are optimized using AdamW [43] with a weight decay of 0 and betas $(0.9, 0.999)$. We use a constant learning rate of $1 \times 10^{-4}$, a batch size of 64, and train for 1.1 million steps in the main experiments (Table 1 and Figure 5). Data augmentation follows the same protocol as Diff-Foley [9]. We do not tune learning rates, apply warm-up or decay schedules, modify AdamW parameters, add extra augmentations, or apply gradient clipping. For ablation studies in Sections 4.3.1–4.3.5, models are trained for 300,000 steps with a batch size of 16, while keeping all other settings unchanged.

**Sampling Configurations.** We utilize an exponential moving average (EMA) of model weights with a decay factor of 0.9999 and employ EMA checkpoints for all sampling, which consistently achieving improved performance. By default, sampling is performed using the Euler-Maruyama solver with 50 denoising steps and a classifier-free guidance (CFG) value of 1.45. For the comparative analysis presented in Figure 5 of the main paper, we adopt the Euler solver with a reduced step count of 25, aligning our methodology with that of FRIEREN [13] to ensure a fair and accurate comparison.

**Metric Calculation** We utilize audio evaluation tools provided by AudioLDM [37] for FAD, FD, IS, and KL. For alignment accuracy, we use the code provided by Diff-Foley [9].

**Architectural Configurations.** We adopt the same transformer architecture as SiT [35], experimenting with four model scales: MGAudio-S, B, L, XL, varying in parameter count and computational cost. A detailed summary is provided in Tab. 8.

## C  Learned Data Distribution Comparison

We present a higher-resolution comparison of the generated and real audio sample distributions in Fig. 7. For each audio sample, both generated and real, we first extract sample-wise logits using

Table 8: **Transformer Model Configurations of MGAudio with different model size.**

| Model | Layer N | Hidden dimension $d$ | Head | Patch size | # Parameters (M) |
|---|---|---|---|---|---|
| MGAudio-S | 12 | 384 | 6 | 2 | 34 |
| MGAudio-B | 12 | 768 | 12 | 2 | 131 |
| MGAudio-L | 24 | 1024 | 16 | 2 | 464 |
| MGAudio-XL | 28 | 1152 | 16 | 2 | 680 |

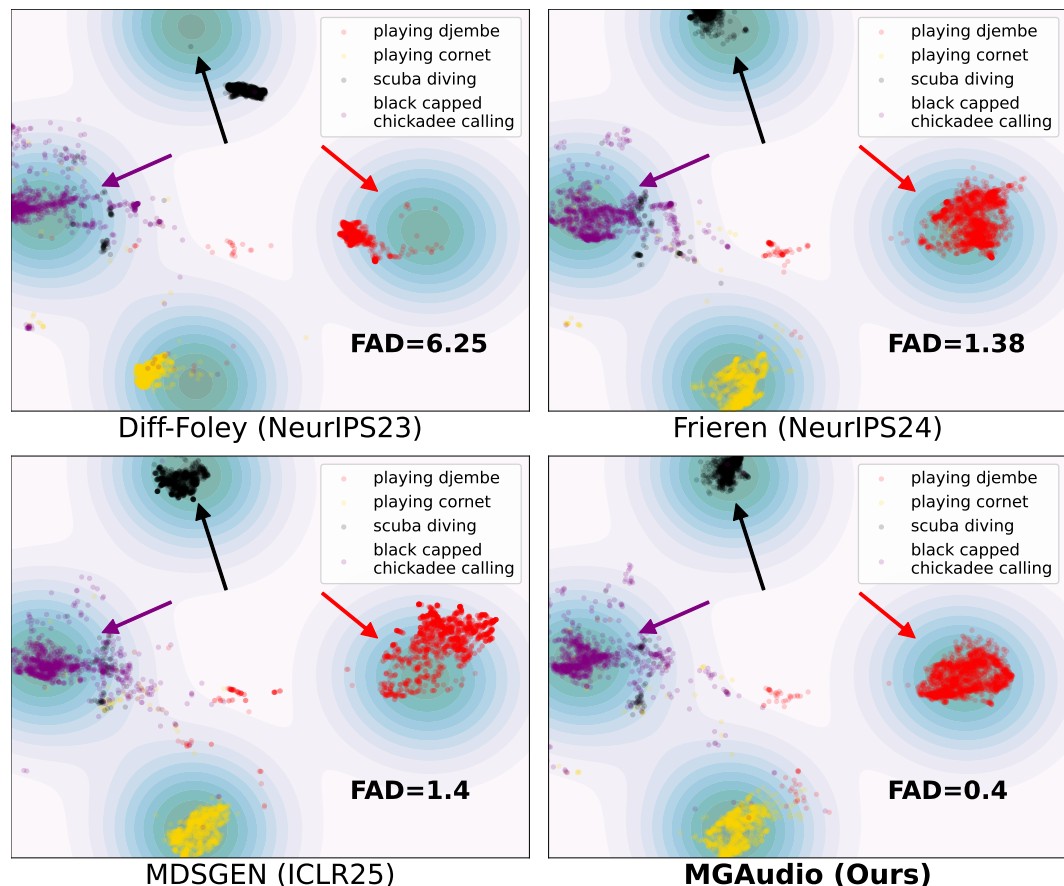

Figure 7: **Audio Distribution.** `MGAudio` generates audio samples that *more closely align* with the target distribution compared to other methods. For example, samples for classes "*playing djembe*" (red points) and "*scuba diving*" (black points) are tightly clustered around the center of the real sample distribution.

the PANNs model [44]. These logits are then projected into a 2D space using UMAP [39] for visualization. The resulting embeddings are used to plot the distribution of generated samples, along with contour plots of the real samples. The filled contours in Fig. 7 represent the density of real audio samples from VGGSound, focusing on four selected classes: *playing djembe*, *playing cornet*, *scuba diving*, and *black-capped chickadee calling*.

# D   Effect of Batch Size

We analyze the influence of varying batch sizes on the stability and quality of MGAudio training. As shown in Tab. 9, larger batch sizes generally improve key performance metrics, particularly FAD, FD, IS, and alignment accuracy, suggesting enhanced audio fidelity, diversity, and alignment quality.

For our main experiments, we select a batch size of 64, which yields the best overall performance metrics across all evaluated criteria. For computational efficiency in our extensive ablation studies, we opt for a smaller batch size of 16, which provides reasonable performance metrics despite some degradation, enabling faster experimentation cycles.

Table 9: Impact of Batch Size on MGAudio Performance Metrics

| Batch Size | FAD↓ | FD↓ | IS↑ | KL↓ | Align. Acc. |
|---|---|---|---|---|---|
| 16 | 1.14 | 13.09 | 8.35 | **2.70** | 93.67 |
| 32 | 0.71 | 12.07 | 8.61 | 2.72 | 94.41 |
| **64** | **0.51** | **9.19** | **10.25** | 2.75 | **95.14** |

# E    Effect of Sampling Step

We evaluate the performance of MGAudio under different sampling step budgets: 5, 25, 50, 120, 250, and 500 steps. As shown in Fig. 8, MGAudio achieves optimal performance for major metrics such as FAD, FD, and KL at 25-50 sampling steps. This indicates that further increasing sampling steps does not necessarily enhance results and may incur unnecessary computational overhead.

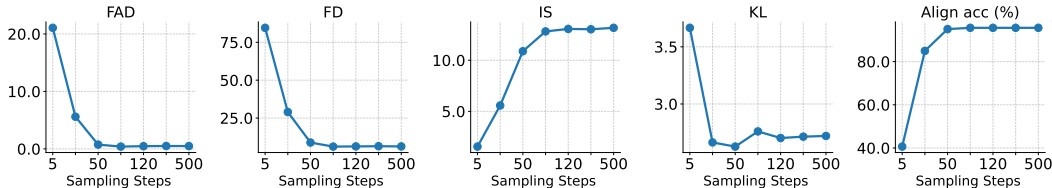

Figure 8: **Performance Metrics of MGAudio Across Different Sampling Steps.**

Furthermore, to ensure a fair comparison across methods, we evaluate all models using identical sampling steps or number of forward evaluation (NFE) while adopting the default CFG value reported in each method. As shown in Table 10, our approach achieves competitive or superior results under these standardized and transparent settings. Notably, even at ( NFE = 100 ), MGAudio maintains inference times comparable to FRIEREN and MDSGen while being substantially faster than MMAudio. Moreover, at ( CFG = 1.0 ), our model surpasses both FRIEREN and MDSGen in overall quality with only 50 NFEs, demonstrating its efficiency and strong generation capability.

Table 10: Comparison of inference efficiency and quality under different CFG values and sampling settings.

| Method | CFG | Steps | NFE | FAD↓ | FD↓ | IS↑ | Time (s)↓ |
|---|---|---|---|---|---|---|---|
| FRIEREN | 4.50 | 25 | 50 | 1.38 | 12.36 | 12.12 | 0.20 |
| MDSGen | 5.00 | 25 | 50 | 1.40 | 17.42 | 9.66 | 0.24 |
| MMAudio | 4.50 | 25 | 50 | 0.71 | 6.97 | 11.09 | 0.98 |
| MGAudio (Ours) | 1.00 | 25 | 25 | 1.40 | 9.04 | 9.19 | **0.08** |
| MGAudio (Ours) | 1.45 | 25 | 50 | 0.68 | 7.49 | 12.11 | 0.15 |
| MGAudio (Ours) | 1.00 | 50 | 50 | 0.80 | 7.89 | 9.90 | 0.15 |
| MGAudio (Ours) | 1.45 | 50 | 100 | **0.40** | **6.16** | **12.82** | 0.31 |

# F    Effect of Longer Training

To study the effect of extended training, we continue training `MGAudio` up to 1.7M steps. As shown in Fig. 9, the model continues to improve steadily with longer training, with performance metrics stabilizing beyond 1.1M steps. We select the 1.1 M-step checkpoint as our final model to balance training efficiency and generation quality.

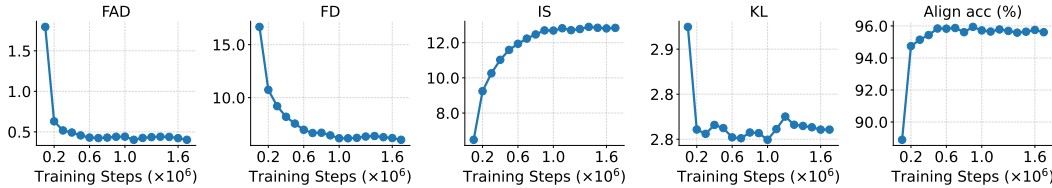

Figure 9: **Longer Training Effect.** We plot the evolution of five evaluation metrics, FAD, FD, IS, KL divergence, and alignment accuracy, over training steps. `MGAudio` exhibits consistent improvements across all metrics during the first 1M steps, after which most metrics begin to saturate. Notably, FAD and FD steadily decrease, and IS and alignment accuracy improve, indicating that both perceptual quality and semantic consistency benefit from prolonged training.

Interestingly, prior work based on diffusion models such as MDSGen [12] reports that longer training can lead to degraded performance, likely due to overfitting or mode collapse. In contrast, our method maintains or improves quality throughout, highlighting its robustness and the effectiveness of the model-guided training objective with flow-matching learning.

## G   Robust Learning from Noisy or Limited Data

We observe that MGAudio maintains strong performance even when trained with only 10% of the VGGSound dataset, suggesting improved robustness to limited and potentially noisy training data. This robust learning capability under constrained data conditions has not been observed in prior Vision Model Guidance [16].

To verify this observation, we train MGAudio (with and without AMG loss) and MMAudio using the same 10% subset of VGGSound under identical settings (1M steps). As shown in Table 11, MGAudio with the AMG loss substantially outperforms both its ablation and MMAudio across all metrics, indicating that the AMG loss may promote more stable and data-efficient learning.

Table 11: **Performance Comparison when models train with 10% of VGGSound dataset.**

| Method | AMG Loss | FAD↓ | FD↓ | IS↑ | KL↓ |
|---|---|---|---|---|---|
| MMAudio [14] | - | 2.81 | 14.28 | 10.37 | 2.87 |
| MGAudio | × | 32.84 | 123.14 | 1.09 | 4.34 |
| MGAudio | ✓ | **0.82** | **8.13** | **10.68** | **2.67** |

Mechanistically, we hypothesize that AMG's self-distilled flow targets act as a dynamic regularizer: during training, the model infuses its own predicted flow (conditioned on the visual cue) into the target flow, effectively "rectifying" noisy or weak samples. This process stabilizes learning and yields stronger generalization under data scarcity.

## H   Effect of Representation Alignment Weight $\lambda$

We investigate the effect of varying the weight $\lambda$ of the audio representation alignment loss, following the setup in prior work [25]. As shown in Table 12, the model achieves the best perceptual quality (lowest FAD) when $\lambda = 0.5$. Increasing $\lambda$ beyond this value slightly improves FD and KL but leads to a degradation in perceptual realism. Overall, $\lambda = 0.5$ provides a balanced trade-off between perceptual quality and distributional fidelity, and the model remains stable across a reasonable range of $\lambda$. Therefore, we adopt $\lambda = 0.5$ as the default setting in all subsequent experiments.

## I   Sampler Investigation

We investigate two widely-used sampler families: deterministic ODE solvers (Euler integrator) and stochastic SDE samplers (Euler-Maruyama integrator). Following prior works [35, 16], both samplers

Table 12: **Ablation study on $\lambda$, the weight of the audio representation alignment loss.**

| Metric | $\lambda = 0.25$ | $\lambda = 0.5$ | $\lambda = 0.75$ | $\lambda = 1.0$ |
|---|---|---|---|---|
| FAD↓ | 0.74 | **0.71** | 0.78 | 0.81 |
| FD↓ | 13.48 | 12.07 | 11.69 | **11.64** |
| KL↓ | 7.75 | 8.61 | **8.62** | **8.62** |

are evaluated using a consistent 50-step sampling scheme. As presented in Tab. 13, the SDE-based Euler-Maruyama sampler achieves better overall performance, indicating superior audio fidelity and diversity compared to the deterministic Euler sampler. Thus, for our main experiments, we utilize the Euler-Maruyama integrator due to its optimal balance between realism and variation.

Table 13: Performance Comparison between ODE Euler and SDE Euler-Maruyama Samplers.

| Sampler | Type | FAD↓ | FD↓ | IS↑ | KL↓ | Align. Acc. (%) |
|---|---|---|---|---|---|---|
| Euler | ODE | 0.58 | 6.71 | 12.47 | **2.69** | **95.98** |
| Euler-Maruyama | SDE | **0.40** | **6.16** | **12.82** | 2.76 | 95.65 |

## J Video Conditioning Integration Styles

Our default conditioning strategy follows MDSGen [12], which aggregates video frames into a single global feature vector and modulates the audio generator via AdaIN. This approach is computationally efficient but may compromise temporal precision. Although temporal modeling is not the primary focus of this work, we explore alternative conditioning strategies to examine their potential impact on generation quality.

To ensure a fair comparison, we integrate each conditioning strategy into our framework by selectively enabling its corresponding temporal alignment mechanism while removing others. All models are trained under identical configurations (batch size of 64, 300k iterations) to isolate the effect of each conditioning design. The core part for each strategy is:

- **MDSGen-style (ours):** Global frame aggregation with AdaIN modulation.
- **FRIEREN-style [13]:** Temporal interpolation of frame features to match the mel-spectrogram length, followed by channel-wise concatenation.
- **MMAudio-style** [14]: Multimodal transformer with joint audio–video attention.

We evaluate each strategy using MMAudio's metric suite, which covers distributional fidelity (FD-VGG, FD-PASST), semantic consistency (IB-Score), and temporal alignment (DeSync):

Table 14: **Comparison of video conditioning strategies.** Lower is better for FD and DeSync; higher is better for IB-Score.

| Conditioning Style | FD-VGG ↓ | FD-PASST ↓ | IB-Score ↑ | DeSync ↓ |
|---|---|---|---|---|
| MDSGen-style | **0.5411** | **74.092** | 24.58 | 1.242 |
| FRIEREN-style | 0.8293 | 90.668 | 23.56 | **0.938** |
| MMAudio-style | 0.6327 | 81.732 | **25.05** | 0.961 |

**Findings.** As shown in Table 14, we have the following findings: (1) MDSGen-style conditioning achieves the best distributional fidelity, likely due to its compact and stable global representation. (2) FRIEREN-style provides the strongest temporal alignment, benefiting from explicit frame-level interpolation. (3) MMAudio-style yields the highest semantic consistency, owing to its cross-modal attention design.

These results suggest that temporal conditioning design is largely orthogonal to our core contributions—AMG and dual-role alignment. We adopt the MDSGen-style configuration for its simplicity

and strong overall performance, but acknowledge that incorporating finer temporal granularity remains a promising direction for future work.

Table 15: **Training Time for Different MGAudio Model Sizes.** Wall-clock training durations on a single A6000 GPU for 300,000 iterations using mixed-precision training and a batch size of 16.

| Model | Training Time (hours) |
|---|---|
| MGAudio-S | 16.6 |
| MGAudio-B | 25.7 |
| MGAudio-L | 57.0 |
| MGAudio-XL | 77.1 |

# K   Training Time Analysis

Tab. 15 details the training durations for various MGAudio model sizes (small to extra-large). Training was conducted on a single A6000 GPU using mixed-precision optimization for 300,000 iterations with a batch size of 16. We select MGAudio-B as our primary model due to its balanced trade-off between performance and computational efficiency. The results highlight the increasing computational requirements associated with larger model architectures, illustrating practical considerations in choosing model complexity.

# L   Additional Metrics Comparison

To comprehensively evaluate audio quality beyond standard metrics such as FAD and FD, we follow [14] and include additional metrics shown in Table 16. These metrics encompass Fréchet Distance (FD) computed using multiple audio models (VGG [45], PANNs [44], PaSST [46]), and Kullback-Leibler divergence (KL) evaluated using PANNs and PaSST embeddings for distribution matching between generated audio and ground truth audio. And Inception Score (IS) [47] to assess audio quality. Semantic alignment is evaluated using IB-Score, where ImageBind [48] extracts visual and audio features, computing average cosine similarity as in [49]. Audio-visual synchrony is measured via DeSync scores using Synchformer to estimate temporal alignment errors.

As detailed in Table 16, MGAudio consistently outperforms competing methods in key distribution metrics (FD-VGG and FD-PANNs) and achieves the highest audio quality according to IS with a CFG of 6.0. Additionally, despite MMAudio achieving the lowest DeSync, MGAudio demonstrates a strong semantic alignment (IB-Score), particularly at higher CFG values, without the need for explicit test-time optimization as done by See and Hear. Furthermore, MGAudio accomplishes these superior results with fewer parameters (131M) compared to MMAudio (157M), underscoring its efficiency and effectiveness.

Table 16: **Metrics from MMAudio [14].** `MGAudio` **on VGGSound.** Audio generation quality across state-of-the-art methods on the VGGSound test set is presented. Bold indicates the best performance, and an underline denotes the second-best. *We train MMAudio from scratch solely on the VGGSound dataset, and set the text input to *None* during inference for fairness.

| Method | FD↓ | | | KL↓ | | IS↑ | IB-Score↑ (s) | # DeSync↓ | # Params↓ |
|---|---|---|---|---|---|---|---|---|---|
| | VGG | PANNs | PaSST | PANNs | PaSST | PANNs | | | |
| Diff-Foley [NeurIPS'23] [9] | 6.25 | 23.07 | 358.92 | 3.17 | 3.04 | 10.77 | 19.88 | 0.91 | 860M |
| See and Hear [CVPR'24] [10] | 5.51 | 26.60 | 227.67 | 2.82 | 2.78 | 5.71 | **36.11** | 1.20 | 1099M |
| FRIEREN [NeurIPS'24] [13] | 1.38 | 12.36 | 107.57 | 2.72 | 2.84 | 12.12 | 22.83 | 0.85 | 157M |
| MDSGen [ICLR'25] [12] | 1.40 | 17.42 | 114.27 | 2.83 | 2.80 | 9.68 | 22.53 | 1.23 | **131M** |
| MMAudio [CVPR'25]* [14] | 0.71 | 6.97 | **51.00** | **2.08** | **1.97** | 11.08 | 27.35 | **0.50** | 157M |
| **MGAudio**, CFG = 1.45 | **0.40** | **6.16** | 83.53 | 2.75 | 2.56 | 12.80 | 26.53 | 1.22 | **131M** |
| **MGAudio**, CFG = 4.0 | 2.23 | 13.65 | 77.41 | 2.75 | 2.62 | 14.41 | 27.13 | 1.23 | **131M** |
| **MGAudio**, CFG = 6.0 | 1.50 | 9.10 | 69.75 | 2.81 | 2.62 | **17.36** | 28.77 | 1.21 | **131M** |

# M   Human Preference Evaluation

We conducted a human evaluation to compare MGAudio with two baselines (FRIEREN [13] and MMAudio [14]) using a 5-point Likert scale. Three participants were asked to first watch the silent video and then listen to the generated audio clips from each method without being informed of their source. Participants rated each clip from 1 to 5, where '1" indicates poor audiovisual alignment and unnatural sound, and '5" indicates strong synchronization and natural acoustic quality.

As shown in Table 17, listeners consistently preferred MGAudio over both FRIEREN and MMAudio, indicating that our method produces audio that better aligns with visual events and is more perceptually natural.

Table 17: Average human preference scores (1–5) for generated audio across different methods.

| Method | FRIEREN | MMAudio | MGAudio (ours) |
|---|---|---|---|
| Preference | $3.44 \pm 0.46$ | $3.89 \pm 0.47$ | **$4.38 \pm 0.39$** |

# N   Qualitative Comparisons

We present qualitative comparisons between `MGAudio` and prior state-of-the-art methods on randomly selected videos from the VGGSound test set. Each example shows input video frames followed by the mel-spectrograms generated by different models. As illustrated in Fig. 10–13, `MGAudio` produces spectrograms with more realistic structure, temporal continuity, and acoustic richness, closely resembling the ground truth. Corresponding audio samples are included in the supplementary materials. These results highlight the benefits of the model-guided objective in generating coherent and semantically accurate audio. Beside the samples show at here, we have included more samples generated from different method in the supplementary material.

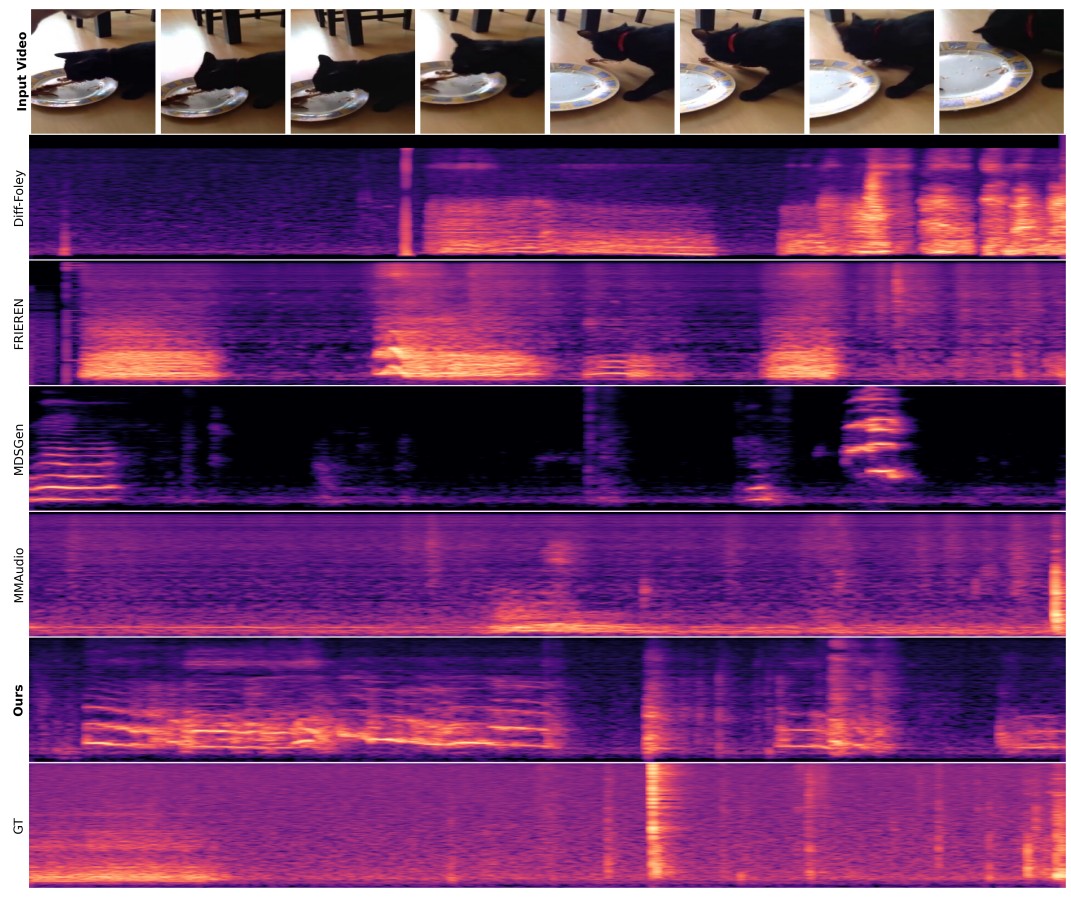

Figure 10: **Qualitative Comparison of Mel-Spectrograms.** The example is from the VGGSound test set (video: "A8rklgn3N4A_000028.mp4"), depicting a cat growling.

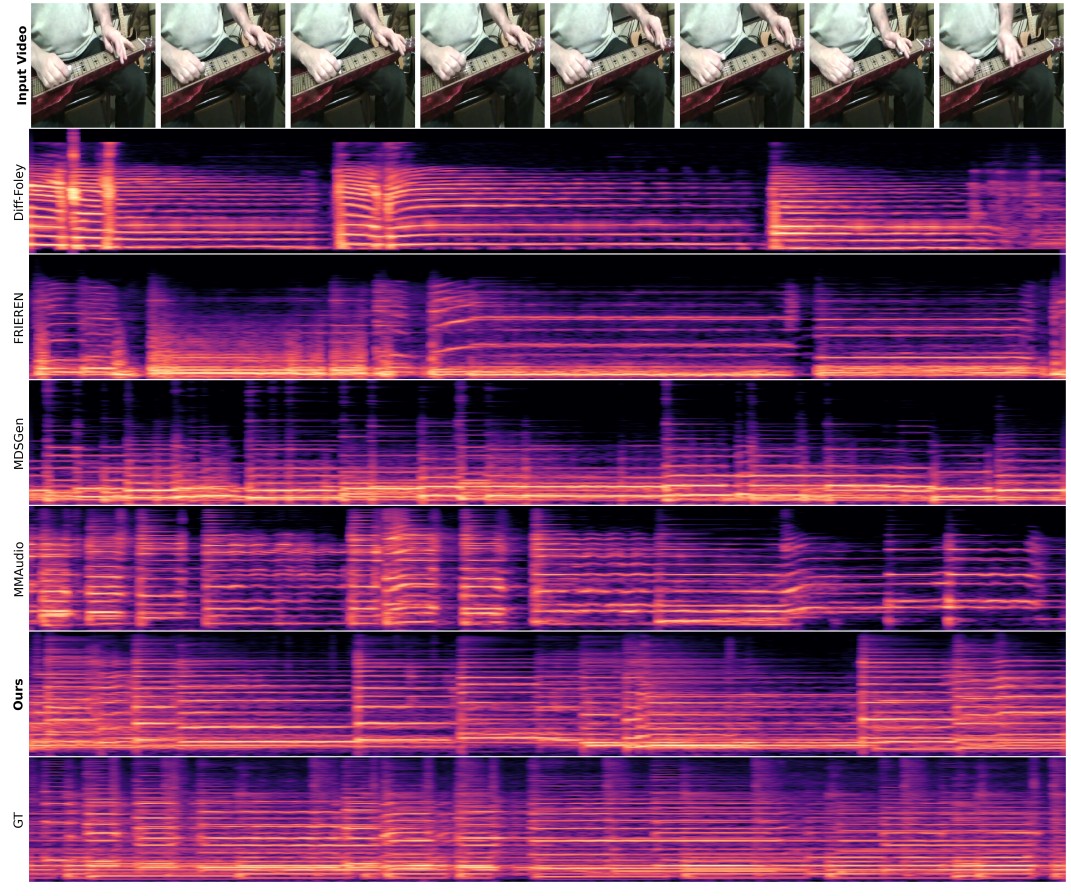

Figure 11: **Qualitative Comparison of Mel-Spectrograms.** The example is from the VGGSound test set (video: "GO2Tf8KLJ14_000061.mp4"), depicting a person playing steel guitar, slide guitar.

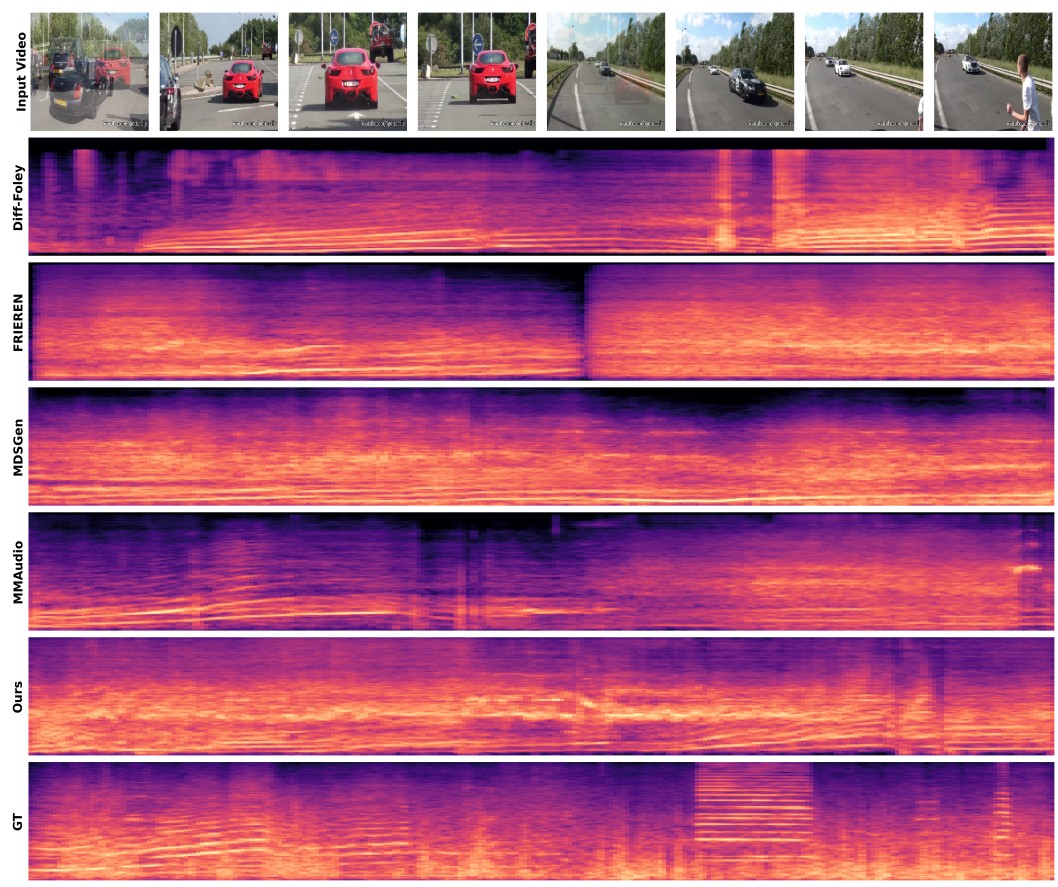

Figure 12: **Qualitative Comparison of Mel-Spectrograms.** The example is from the VGGSound test set (video: "hHbJRPjqgXQ_000090.mp4"), depicting a race car, auto racing.

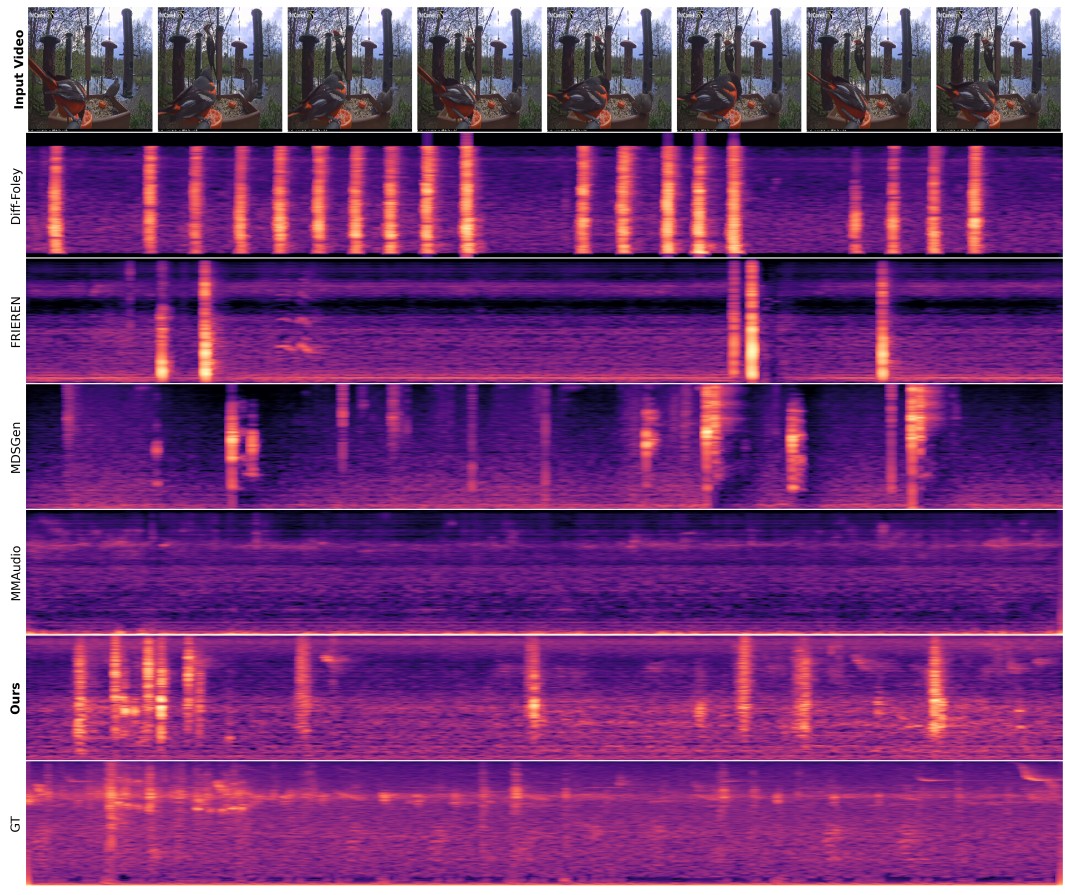

Figure 13: **Qualitative Comparison of Mel-Spectrograms.** The example is from the VGGSound test set (video: "mKEJRZtNx9o_000044.mp4"), depicting a baltimore oriole calling his mate.

