# OpenReview forum: "Model-Guided Dual-Role Alignment for High-Fidelity Open-Domain Video-to-Audio Generation"
_NeurIPS.cc/2025/Conference — NeurIPS 2025 poster_

### Official Review · Reviewer_3BgF · 2025-06-30

**Clarity:** 3
**Significance:** 2
**Originality:** 2
**Rating:** 5
**Confidence:** 4

**Summary:**

This paper introduces MGAudio, which incorporates scalable interpolant transformers (SiT), model guidance, and representation alignment into the video-to-audio task. When trained on VGGSound, MGAudio improves over prior methods on both seen and unseen data.

**Questions:**

The choice of a Mel spectrogram representation for audio is probably suboptimal relative to the embedding space of recent audio encoders. Mel spectrograms are relatively uncompressed, contain many low-energy bins, and discard phase information. Have you ablated this choice of input representation with recent audio encoders?

What I am most concerned about is the data efficiency experiment. Less training data could perform better due to the fewer training steps; in that case, the model may have seen each example an optimal number of times. The data efficiency experiment is currently non-conclusive without also showing results at the full number of training steps—or, better yet, find the optimal number of steps for each dataset size.

**Ethical Concerns:**

["NO or VERY MINOR ethics concerns only"]

**Final Justification:**

Authors address my primary concern of the data efficacy potentially not being robust with a small experiment that further demonstrates robustness and another that isolates the key component responsible. Further work by the authors in ablations and subjective evaluations improves an already well-written paper. The methods employed are not novel, but they are very recent methods being successfully applied for the first time to new domains.

**Limitations:**

Yes

**Quality:**

3

**Strengths And Weaknesses:**

This paper corroborates the efficacy of three recent methods (SiT, model guidance, and representation alignment) in the video-to-audio domain. The methods the authors chose are of significant recent interest to the community, demonstrating broad potential for improvement. Here, that improvement is concretely demonstrated for video-to-audio and any additional challenges for adaptation are discussed. However, the three methods are not original in themselves; this paper serves more as a useful validation that the methods are worth using. To that end, it is a clear, well-organized, and polished paper.

The evaluation results on VGGSound (Table 1) are good, but incremental. The generalization results on UnAV-100 are a more convincing argument of the efficacy of the proposed MGAudio (Table 2). The extensive evaluations afterwards that discuss potential reasons for the observed improvement are well done.

---

> ### Author Rebuttal · Authors · 2025-07-30
>
> We thank the reviewer for recognizing our integration of recent techniques into the MGAudio framework and for highlighting our model’s generalization to unseen data. Below, we address your questions.
>
> ---
>
> > 1. **Questions 1:** The choice of a Mel spectrogram representation for audio is probably suboptimal relative to the embedding space of recent audio encoders. Mel spectrograms are relatively uncompressed, contain many low-energy bins, and discard phase information. Have you ablated this choice of input representation with recent audio encoders?
>
> We appreciate this insightful suggestion. We chose a VAE‑based Mel‑spectrogram latent to maintain comparability with prior work, but agree that more modern audio encoders (e.g., waveform‑level encoder) might offer superior compression and phase preservation. Importantly, our dual‑role alignment and Model Guidance (MG) training objective are encoder‑agnostic. In future work, we will evaluate MGAudio with state‑of‑the‑art audio encoders and expect further gains by directly modeling waveform embeddings.
>
> ---
>
> > 2. **Questions 2:** What I am most concerned about is the data efficiency experiment. Less training data could perform better due to the fewer training steps; in that case, the model may have seen each example an optimal number of times. The data efficiency experiment is currently non-conclusive without also showing results at the full number of training steps—or, better yet, find the optimal number of steps for each dataset size.
>
> Thank you for pointing out this potential confound. To isolate the effect of the AMG loss, we conducted a controlled experiment both with and without AMG loss for 1 million steps on randomly selected 10% subset of VGGSound. We also train MMAudio on the same subset of VGGSound. The results are shown as follows:
>
> **Table 1:** Comparison of fully trained MGAudio (with and without AMG) and MMAudio on the same randomly selected 10% subset of VGGSound.
> | Method  | AMG Loss | FAD ↓ | FD ↓   | IS ↑  | KL ↓ |
> |---------|----------|-------|--------|-------|------|
> | MGAudio (10%) | ✓        | 0.82  | 8.13   | 10.68 | 2.67 |
> | MGAudio (10%) | ✗        | 32.84 | 123.14 | 1.09  | 4.34 |
> | MMAudio (10%) | -        | 2.81  | 14.28  | 10.37 | 2.87 |
>
> These results confirm that (1) removing AMG dramatically degrades all metrics, even when both models train for the same number of steps and same dataset; and (2) our MGAudio with AMG outperforms the prior MMAudio baseline in limited data scenarios. This supports our claim that AMG has strong learning capability with scarce data.
>
> Mechanistically, we hypothesize that AMG’s self‑distilled flow targets act as a dynamic regularizer: during training, the model infuses its own predicted flow (conditioned on the visual cue) into the target, effectively “rectifying” noisy or weak samples. This process stabilizes learning and yields stronger generalization under data scarcity.

---

> > ### Comment · Reviewer_3BgF · 2025-08-03
> >
> > The authors have sufficiently addressed my concerns in their rebuttal. Additional rebuttals include subjective evaluation results and temporal alignment experiments that are a welcome addition—and further convince me that the methods utilized by the authors are worth boosting in visibility. I am raising my score from a four to a five.

---

> ### Author Response · Authors · 2025-08-04
>
> **We sincerely thank the reviewer for the thoughtful feedback and for raising the score.** We're glad that the additional subjective evaluation results and temporal alignment experiments helped clarify our contributions. Your recognition of the value in our proposed methods is truly appreciated, and we are encouraged by your support to further develop and refine this line of research.

---

### Official Review · Reviewer_yNJc · 2025-07-01

**Clarity:** 3
**Significance:** 3
**Originality:** 2
**Rating:** 5
**Confidence:** 4

**Summary:**

The paper presents a flow-based framework, named MGAudio, for open-domain video-to-audio (V2A) generation. The main idea is using model-guided dual role alignment to allow the generative model to guide itself through a dedicated training objective, unlike previous methods that rely on CFG. The framework integrates three main components: 1) a scalable flow-based transformer denoiser that is based on SiT architecture and learns continous transport directions, 2) a dual-role alignment mechanism where the audio-visual encoder serves both as a conditioning module and as a feature aligner, and 3) a model-guided objective (Audio Model-Guidance) that replaces CFG by using a self-distilled target to refine model predictions. MGAudio achieves SOTA performance on VGGSound and demonstrates good generalization to UnAV-100 benchmark.

**Questions:**

Besides clarification questions mentioned, I am also interested in the following:
- Could authors provide a more detailed hypothesis for why the audio domain uniquely benefits from AMG training + CFG inference? What specific characteristics of audio, such as its temporal sensitivity, its often ambiguous mapping to visual cues or something else, will require this combined approach?
- How does MGAudio's Audio Model-Guidance (AMG) objective specifically contribute to its robustness against noisy/irrelevant data within VGGSound, beyond merely learning efficiently from less data? Does its "self-distilled target" implicitly down-weight or ignore poor samples? Could the authors propose specific strategies for future work that could explicitly leverage AMG's properties to actively identify, filter, or mitigate the impact of low-quality audio-visual pairs during training, further enhancing performance and data efficiency?

**Ethical Concerns:**

["NO or VERY MINOR ethics concerns only"]

**Final Justification:**

the rebuttal addressed most of my concerns and answered all my questions. While several established components were used in this work, I agree that the overall design is non-trivial. With the new subjective eval result and clarifications presented in the rebuttal, this is indeed a solid application paper.

**Limitations:**

yes

**Quality:**

3

**Strengths And Weaknesses:**

**Strengths**
- MGAudio gets SOTA performance on VGGSound (FAD = 0.4) and outperforms previous CFG baselines.
- The model demonstrates good generalization to the UnAV-100 benchmark
- Training on 10% VGGSound dataset still achieves good results. It seems like Audio Model-Guidance does enhance efficient learning
- Extensive ablation studies are available in both main paper and appendix. The detailed implementation setup is also healthy for reproducibility.
- The motivation and components in MGAudio framework are clear. Overall I think the model-guided dual-role alignment is a cool idea and makes sense to me. The paper is generally easy to understand.

**Weaknesses**
- For video-to-audio generation, it is quite important to have a human evaluation. While I really want to believe the MGAudio framework is as good as shown in all quantitative metrics numbers, I was not fully convinced by watching video samples provided in the supplementary material. Therefore, I will recommend authors adding one human study to get the perceptual difference.
- While the paper presents new adaptations and a coherent combination of various techniques (SiT, CAVP, Model-Guided framework), there is relatively few new concept introduced in this paper.
- As authors mentioned in the paper, the current framework would not handle human vocalizations well and the model-guided training may introduce confusion when visual semantics are ambiguous or loosely correlated with the target sound.
- Several typos and clarification:
  - L131-137: please use accurate notations and clarify all numbers mean even though some of them are straight-forward to guess (e.g., 3x224x224 are frames but what about 64 and 816 in $X$ ?).
  - L134: pathchified -> patchified
  - Section 4.3.1: please clarify how 10/30/50% of the training data are being selected. I assume you were selecting them from a uniform distribution. Please also clarify how you define weak or irrelevant audio-visual alignment during manual inspection of the 100 random samples.
  - Section 4.3.5: Please clarify why the four distinct VGGSound classes are selected (why not other classes?). And also clarify why Figure 4 could tell us the generate sample has stronger diversity.

---

> ### Author Rebuttal · Authors · 2025-07-30
>
> We thank the reviewer for their thoughtful and detailed feedback. We address each concern below and will incorporate the necessary clarifications in the final version.
>
> ---
>
> > 1. **Weaknesses 1:** For video-to-audio generation, it is quite important to have a human evaluation. While I really want to believe the MGAudio framework is as good as shown in all quantitative metrics numbers, I was not fully convinced by watching video samples provided in the supplementary material. Therefore, I will recommend authors adding one human study to get the perceptual difference.
>
> Thank you for emphasizing the importance of perceptual evaluation. We fully agree that human judgment is essential for assessing video-to-audio quality. We have conducted a human study where participants are asked to rate audio samples (MGAudio vs. baselines) based on overall preference using a 5-point Likert scale. Below is a summary of the results:
>
> **Table 1:** Overall user preference for generated audio across different methods.
> | Method     | FRIEREN | MMAudio | MGAudio (ours) |
> |------------|---------|---------|----------------|
> | Preference |  3.44 ± 0.46  |   3.89 ± 0.47  |   4.38 ± 0.39    |
>
> Listeners consistently preferred MGAudio. We will include full study details in the appendix and summarize key findings in the main paper.
>
> ---
>
> > 2. **Weaknesses 2:** While the paper presents new adaptations and a coherent combination of various techniques (SiT, CAVP, Model-Guided framework), there is relatively few new concept introduced in this paper.
>
> While we do build upon established components (e.g., SiT, CAVP, model guidance), our contribution lies in the **novel role assignment and integration of multimodal representations** within a **dual-role alignment framework**. Specifically:
>
> * We leverage **CAVP**, a pretrained audio-video-aligned encoder, in two distinct and complementary ways:
>
>   1. **Audio Model Guidance (AMG):** Uses the **video representation** to guide the training through a CFG-like objective on the target flow.
>   2. **Representation Alignment (REPA):** Uses the **audio representation** to regularize the internal latent space of the generative model.
>
> * This **dual use of a single pretrained encoder**—as both a training signal and a regularizer—has not been explored in prior V2A works. Most prior approaches either treat video features merely as conditional inputs or impose no constraint on the model’s internal representations.
>
> * Our experiments show that this synergy significantly improves **semantic relevance**, **temporal synchronization**, and **perceptual quality**, while maintaining high **data efficiency**.
>
> This design is **non-trivial**: training a strong multi-modal representation model is computationally expensive, and naively using it as a condition vector does not leverage its full potential. We show that learned representation specifically for the target generating modality and guiding modality can be used to guide generation through the combination of model guidance (MG) and representation alignment (REPA). This presents a **promising, general framework for future cross-modal generation tasks**: first, train a powerful multi-modal representation learner, then leverage it for both trajectory guidance and generative modal internal feature alignment.
>
>
> ---
>
> > 3. **Weaknesses 3:** As authors mentioned in the paper, the current framework would not handle human vocalizations well and the model-guided training may introduce confusion when visual semantics are ambiguous or loosely correlated with the target sound.
>
>
> We agree, and appreciate the reviewer’s attention to this limitation. Human vocalization remains challenging for most open-domain video-to-audio models, including FRIEREN, MDSGen, and MMAudio. One key reason is the lack of rich phonetic structure and fine-grained facial cues in the VGGSound dataset, coupled with a limited amount of speech-related data. These factors hinder accurate synthesis of vocal content.
>
> Furthermore, when visual cues are ambiguous (e.g., static scenes, occlusions, or off-screen sounds), it becomes difficult to disambiguate the correct audio modality. Addressing these issues, for instance, by incorporating visual speech encoders or training on speech-rich datasets, is an important future direction.
>
> ---
>
> > 4. **Weaknesses 4:** Several typos and clarification:
> > - L131-137: please use accurate notations and clarify all numbers mean even though some of them are straight-forward to guess (e.g., 3x224x224 are frames but what about 64 and 816 in X?).
> > - L134: pathchified -> patchified
> > - Section 4.3.1: please clarify how 10/30/50% of the training data are being selected. I assume you were selecting them from a uniform distribution. Please also clarify how you define weak or irrelevant audio-visual alignment during manual inspection of the 100 random samples.
> > - Section 4.3.5: Please clarify why the four distinct VGGSound classes are selected (why not other classes?). And also clarify why Figure 4 could tell us the generate sample has stronger diversity.
>
> * **L131–137**: We clarify that the input audio is represented as a Mel-spectrogram of shape **64 × 816**, where 64 is the number of time frames and 816 is the number of frequency bins.
>
> * **L134**: We will correct “pathchified” to “patchified.”
>
> * **Section 4.3.1**: The 10/30/50% subsets were uniformly sampled from the training set. For manual inspection of 100 samples, “weak or irrelevant alignment” refers to cases where the visual content was insufficient (e.g., static scenes) or misleading relative to the corresponding audio.
>
> * **Section 4.3.5**: We selected four VGGSound classes—*playing djembe, playing cornet, scuba diving, black-capped chickadee calling*—for their strong semantic separability, which improves interpretability of latent clustering. Classes like *laughing* and *screaming* were avoided due to overlap even in ground truth samples. As stated in L322, Figure 4 shows that MGAudio's generated samples closely match the real distribution (semantic accuracy) while exhibiting broader coverage (diversity), which are both desirable traits in generative modeling.
>
> ---
>
> > 5. **Questions 1:** Could authors provide a more detailed hypothesis for why the audio domain uniquely benefits from AMG training + CFG inference? What specific characteristics of audio, such as its temporal sensitivity, its often ambiguous mapping to visual cues or something else, will require this combined approach?
>
> We believe **AMG mitigates the training–inference mismatch** by incorporating CFG-like guidance during training, which enhances compatibility with CFG during inference. This enables the model to benefit from CFG even more effectively than in standard model training. This behavior mirrors findings in the original Model Guidance (MG) paper [16], where MG during training improved both standalone performance and CFG-based inference.
>
> Furthermore, we hypothesize that **audio generation uniquely benefits from AMG + CFG** due to the **high ambiguity of audio given visual input**. Unlike image generation, a single visual scene may correspond to multiple plausible sounds (e.g., walking on grass vs. gravel, or off-screen events). AMG injects a stronger, self-consistent signal during training to resolve such ambiguities. At inference, CFG amplifies correct generation trajectories in low-visual-saliency scenarios, making the two strategies complementary.
>
> We will elaborate further on this point in Section 4.3.6 and the Conclusion.
>
> ---
>
> > 6. **Questions 2:** How does MGAudio's Audio Model-Guidance (AMG) objective specifically contribute to its robustness against noisy/irrelevant data within VGGSound, beyond merely learning efficiently from less data? Does its "self-distilled target" implicitly down-weight or ignore poor samples? Could the authors propose specific strategies for future work that could explicitly leverage AMG's properties to actively identify, filter, or mitigate the impact of low-quality audio-visual pairs during training, further enhancing performance and data efficiency?
>
> Thank you for this insightful question. We believe AMG’s self-distilled targets serve as a **form of regularization**, enabling the model to learn stable representations even in the presence of noisy or weak visual cues. During training, the target flow direction is adjusted using a CFG-like term based on the model’s own prediction, effectively **rectifying implausible training signals** rather than ignoring them.
>
> This mechanism allows AMG to outperform baselines in low-resource and noisy conditions, as demonstrated below:
>
> **Table 2:** Comparison of fully trained MGAudio (with and without AMG) on the same randomly selected 10% subset of VGGSound.
> | AMG Loss  | FAD ↓ | FD ↓   | IS ↑  | KL ↓ |
> |-----------|-------|--------|-------|------|
> | ✓         | 0.82  | 8.13   | 10.68 | 2.67 |
> | ✗         | 32.84 | 123.14 | 1.09  | 4.34 |
>
> In the table above we compare the MGAudio model trained with and without AMG loss on 10% of the VGGSound dataset and keep other setting same with the main experiment. As we can see from the table AMG get significant better result compare with the baseline on all the metrics.
>
> A possible strategy to actively use AMG's properties for data quality control would be to **track the discrepancy between the original and self-guided targets** during training. This discrepancy could serve as a proxy for sample quality: samples with large divergence might indicate noisy or misaligned pairs, which could then be **filtered, down-weighted, or reweighted** dynamically during training.

---

> > ### Comment · Reviewer_yNJc · 2025-08-05
> >
> > Thanks for the new results and clarifications. Please incorporate them in the final version. My concerns were addressed and I am happy to raise my rating to Accept.

---

> > > ### Author Response · Authors · 2025-08-06
> > >
> > > Thank you very much for your encouraging response. We’re glad the new results and clarifications addressed your concerns. We will make sure to incorporate them into the final version as suggested. **We truly appreciate your thoughtful feedback and your decision to raise the rating to Accept.**

---

### Official Review · Reviewer_SNdj · 2025-07-02

**Clarity:** 3
**Significance:** 2
**Originality:** 1
**Rating:** 3
**Confidence:** 5

**Summary:**

Authors proposed MGAudio, a V2A model that adapts model guidance and diffusion feature alignment (i.e REPA) to import V2A performance. The paper focuses on model guidance as an alternative (or complementary) to using CFG during inference.

**Questions:**

- I believe replacing the experiment in Sec. 4.3.6 with the corresponding one in Appendix (MGAudio Vs SiT w/CFG) is better to understand the results, since MGAudio and FRIEREN have different training setup and sampling configurations (e.g 25 vs 50 sampling steps for MGAudio)
- The paper does not discuss the base architecture choices sufficiently. For instance, the author method that to condition on the video inout, they aggregate input frames to a global feature vector that they modulate through AdaIN. How are precise temporal information preserved?
- Few key V2A work are missing from the discussion (e.g [1,2, 3])

[1] AV-Link: Temporally-Aligned Diffusion Features for Cross-Modal Audio-Video Generation
[1] Video-to-Audio Generation with Hidden Alignment
[2] V2A-Mapper: A Lightweight Solution for Vision-to-Audio Generation by Connecting Foundation Models

**Ethical Concerns:**

["NO or VERY MINOR ethics concerns only"]

**Final Justification:**

The main contribution of the paper is applying existing techniques from the T2I domain to the V2A and showing its effectiveness.
After discussing my concerns with the authors, my main concerns regarding the novelty of the proposed method (review q3sa and 3BgF shared the same concern) and the poor qualitative results still remain.


Some of my concerns regarding clarity and model design were resolved. The authors did an extensive rebuttal and provided multiple experiments to strengthen their claims. Therefore, I decided to raise my score to boarderline reject.

**Limitations:**

Yes

**Paper Formatting Concerns:**

No Concerns.

**Quality:**

3

**Strengths And Weaknesses:**

**Strengths**
- The paper is well written and well structured.
- Extensive ablations and experiments are provided in the supplementaries
- The paper explores integrating useful techniques to improve V2A performance.

**Weaknesses**
- One major weakness of the paper is the limited novelty. It explores already proposed techniques (model guidance and feature alignment) for the task of V2A.
- The qualitative results in the supplementary are not convincing. The number of provided samples is small (12 samples total), and the results seems to have significantly weaker temporal alignment than SOTA. For example, I found the bln5A2hopKQ_000092.mp4 sample to have worse temporal alignment than FRIEREN.
- The results of the paper are not conclusive. Table 1 compares with baselines but I could not find any information (even in the supplementaries) about the sampling parameters of these baselines. It is hard to make sense of the results without information about the sampling parameters (e.g how many step / CFG value, etc).
- Table 3 shows that MGAudio achieves the best results with 10% of the data, that raises questions about the model scalability with respect to the data. Also, since no comparison of a baseline trained with the same amount of data is reported, it hard to attribute this to the noisiness of VGGsounds or the model architecture.
- The authors motivate the paper with 1) CFG "may dilute the model's capacity" 2) inference without CFG. Yet the authors (1) (similar to model guidance) train with a 10% drop of caption similar to CFG and (2) report their main experiment (Table 1) with CFG.  The paper misses comparison with baselines to support the superiority of their model over other common method for inference without CFG (e.g guidance distillation)

---

> ### Author Rebuttal · Authors · 2025-07-30
>
> We thank the reviewer for the thoughtful and detailed feedback. We address each concern below and will reflect the necessary clarifications in the final version.
>
> ---
>
> > 1. **Weaknesses 1:** A major weakness is limited novelty, as the paper applies existing techniques (model guidance and feature alignment) to V2A.
>
> Below, we detail the two key novelties of MGAudio:
>
> #### 1. **Dual-Role Alignment of Pretrained Multi-Modal Representations**
>
> Our primary novelty lies in **how we integrate and assign distinct roles to multi-modal representations** from a pretrained encoder to guide both the denoising trajectory and the internal latent space:
> - Audio Model Guidance (AMG): Uses audio-aligned video features to guide denoising via a CFG-like term.
> - Representation Alignment (REPA): Uses video-aligned audio features to constrain the generative model’s latent space.
>
> This dual-role use of a single pretrained multi-modal encoder, for both trajectory guidance and latent alignment, is novel in V2A. Prior work treats video features as static conditions without leveraging them as active training signals or imposing latent constraints. Our approach improves performance and data efficiency, especially in low-resource settings.
>
> This design is **non-trivial**: naively conditioning on pretrained encoder outputs underutilizes their capacity. By explicitly separating guiding (AMG) and aligning (REPA) roles, we robustly steer generation across modalities, offering a general framework for cross-modal generation.
>
> #### 2. **Robust Learning from Noisy or Limited Data**
>
> Moreover, **AMG enables training on low-quality or noisy AV pairs**. This effect of improved robust learning from noisy or limited data has not been shown in the original vision-model-guidance paper [16]. As reviewer yNJc noted, AMG’s self-distilled targets act as a regularizer—adjusting the predicted flow via visual cues during training to rectify, rather than ignore, poor samples. This property could help future work in identifying or mitigating low-quality data, improving robustness and efficiency.
>
>
> ---
>
> > 2. **Weaknesses 2:** The qualitative results in the supplementary are not convincing.
>
> Due to supplementary size limits, we included the maximum allowed 12 samples, selected randomly without cherry-picking, which may result in some weaker examples.
>
> To address this, as suggested by reviewer yNJc, we conducted a human study comparing MGAudio and baselines using a 5-point Likert scale. Results are summarized below:
>
> |Method|FRIEREN|MMAudio|MGAudio (ours)|
> |-|-|-|-|
> |Preference|3.44 ± 0.46|3.89 ± 0.47|4.38 ± 0.39|
>
> Listeners consistently preferred MGAudio. We will include full study details in the appendix and summarize key findings in the main paper.
>
> ---
>
> > 3. **Weaknesses 3:** Missing detailed sampling parameters (e.g., number of steps, CFG value) for compared methods.
>
> We thank the reviewer for pointing this out. We summarize the sampling parameters for all compared methods in the table below. The metric values are also in Table 1 of our Appendix.
>
> |Method|CFG value|Sampling steps|NFE|FAD|FD|IS| Time (s) |
> |-|-|-|-|-|-|-|-|
> |FRIEREN       | 4.50|25|50 | 1.38 | 12.36 | 12.12 | 0.20     |
> |MDSGen        | 5.00|25|50 | 1.40 | 17.42 | 9.66  | 0.24     |
> |MMAudio       | 4.50|25|50 | 0.71 | 6.97  | 11.09 | 0.98     |
> |MGAudio (Ours)| 1.00|25|25 | 1.40 | 9.04  | 9.19  | **0.08**     |
> |MGAudio (Ours)| 1.45|25|50 | 0.68 | 7.49  | 12.11  | 0.15     |
> |MGAudio (Ours)| 1.00|50|50 | 0.80 | 7.89  | 9.90  | 0.15     |
> |MGAudio (Ours)| 1.45|50|100| **0.40** | **6.16**  | **12.82** | 0.31     |
>
> Our model achieves competitive or superior results under fair and transparent settings. Notably:
>
> * Even at NFE = 100, our model's inference time remains comparable to FRIEREN and MDSGen and is significantly faster than MMAudio.
>
> * At CFG = 1.0, our model achieves stronger performance than both FRIEREN and MDSGen, with only 50 NFE.
>
> We will include this table and detailed configuration information in the final version for completeness and reproducibility.
>
> ---
>
> > 4. **Weaknesses 4:** Table 3 shows MGAudio excels with 10% data, but no baseline trained on the same subset is reported, leaving scalability and data noise effects unclear.
>
> We appreciate the reviewer’s concern. To address this, we conducted additional experiments where we train both MGAudio (with and without AMG loss) and MMAudio using the same randomly selected 10% subset of VGGSound, under the same training schedule (1M steps) and other settings unchanged.
>
> We conducted additional experiments training MGAudio (with and without AMG loss) and MMAudio on the same 10% VGGSound subset.
>
>
> | Method  | AMG Loss | FAD ↓ | FD ↓   | IS ↑  | KL ↓ |
> |-|-|-|-|-|-|
> | MGAudio (10%) | ✓        | **0.82**  | **8.13**   | **10.68** | **2.67** |
> | MGAudio (10%) | ✗        | 32.84 | 123.14 | 1.09  | 4.34 |
> | MMAudio (10%) | -        | 2.81  | 14.28  | 10.37 | 2.87 |
>
> Results show **MGAudio with AMG loss outperforms both its ablation and MMAudio**, confirming that improvements stem from AMG’s effective regularization and stable learning, not just data noise.
>
> We will include this table and clarify MGAudio’s strong generalization in low-data regimes, highlighting its value for real-world scenarios with limited high-quality AV data.
>
> ---
>
> > 5. **Weaknesses 5:** The authors motivate the paper with concerns about CFG but still (1) drop conditions during training (like CFG) and (2) report results with CFG. No comparison is made with other CFG-free methods (e.g., guidance distillation).
>
> We clarify our motivation and design rationale below.
>
> Our motivation (L45–46) is to address the **training–inference mismatch in CFG**, where models are trained with dropped conditions but sampled with strong guidance—potentially leading to inconsistency. Our proposed **Audio Model Guidance (AMG)** resolves this by **explicitly optimizing model prediction towards the CFG-like guided trajectory**, ensuring alignment between training and inference.
>
> Although AMG drops 10% of conditions during training, the purpose differs from CFG: these samples are used to compute a **self-distilled guidance target**. This helps the model learn a **consistent denoising direction**, improving stability.
>
> As shown in Figure 5, AMG performs strongly without CFG, and applying the same CFG scale used during training yields further gains—**without introducing mismatch**.
>
>
> ---
>
> > 6. **Questions 1:** Replacing the experiment in Sec. 4.3.6 with the one in the Appendix (MGAudio vs. SiT w/CFG) would better clarify the results, since MGAudio and FRIEREN differ in training and sampling setups.
>
> We agree. Comparing MGAudio and SiT w/CFG, which **share the same training and sampling settings**, offers a more controlled analysis of model-guided training.
>
> To improve clarity, we will replace the current plot with the SiT vs. MGAudio w/CFG comparison in the main paper and move the FRIEREN comparison to the supplementary.
>
> ---
>
> > 7. **Questions 2:** The paper lacks sufficient discussion on base architecture choices. How is temporal information preserved when using global video features and AdaIN?
>
> Our default setup follows MDSGen [12], aggregating video frames into a **global feature vector** and applying AdaIN-based modulation. While efficient, this may limit temporal precision.
>
> To better understand this trade-off, we compared three strategies under the same setup (batch size 64, 300k iterations):
>
> 1. MDSGen-style (ours): Global frame aggregation + AdaIN.
>
> 2. FRIEREN-style: Interpolates video features over time dimension to match the mel-spectrogram resolution and applies channel-wise concatenation.
>
> 3. MMAudio-style: Multimodal transformer with joint audio-video attention.
>
> We evaluated them using MMAudio’s metric suite, that cover distributional (FD-VGG and FD-PASST), semantic (IB-Score), and temporal alignment (DeSync). We implement each of the strategies in our framework and get the results as follows:
>
> | Conditioning Style | FD-VGG ↓   | FD-PASST ↓ | IB-Score ↑ | DeSync ↓  |
> | - | - | - | - | - |
> | MDSGen-style       | **0.5411** | **74.092** | 24.58      | 1.242     |
> | FRIEREN-style      | 0.8293     | 90.668     | 23.56      | **0.938** |
> | MMAudio-style      | 0.6327     | 81.732     | **25.05**  | 0.961     |
>
> Findings:
>
> - MDSGen-style offers the best distributional fidelity, likely due to its compact and stable conditioning signal.
>
> - FRIEREN-style excels at temporal alignment, benefiting from frame-level interpolation.
>
> - MMAudio-style achieves the best semantic alignment, thanks to cross-modal attention.
>
> These results confirm that temporal conditioning is orthogonal to our main contributions—AMG and dual-role alignment. We chose MDSGen-style for its simplicity and strong overall performance but agree temporal granularity is an important design axis. We will include these findings and further discussion in the Appendix.
>
> > 8. **Questions 3:** Few key V2A work are missing from the discussion (e.g [R1, R2, R3])
> > [R1] AV-Link: Temporally-Aligned Diffusion Features for Cross-Modal Audio-Video Generation
> > [R2] Video-to-Audio Generation with Hidden Alignment
> > [R3] V2A-Mapper: A Lightweight Solution for Vision-to-Audio Generation by Connecting Foundation Models
>
> Thank you for highlighting these important works. We will include them in the main paper and related work.
>
> * AV-Link [R1] uses temporally aligned self-attention fusion for bidirectional conditioning, while we focus on explicit alignment at the representation and denoising trajectory levels.
> * Hidden Alignment [R2] improves synchronization via encoder design and augmentations; our method instead applies training-time alignment (REPA) and guided sampling (AMG) to shape denoising with learned multimodal features.
> * V2A-Mapper [R3] maps CLIP to CLAP embeddings for modality bridging, whereas we leverage dual-branch encoders with distinct roles for guidance and alignment.

---

> ### Comment · Reviewer_SNdj · 2025-08-03
>
> I thank the authors for taking the time answering my questions and providing extensive experiments in the rebuttal.
>
> **W1 Novelty**
>
> I am still not convinced by by the novelty of the paper of the major contributions. I disagree with the authors that applying existing and proven diffusion training techniques from text-to-video to video-to-audio is non-trivial. There are no essential novel components specific to V2A introduced by the authors and no significant insights are provided to justify the technical contributions. I find the experiment that the authors can train their model with 10% of the data to be interesting. However, no significant insights are provided on why this happens, is this specific to V2A? and which component contributes to this, whether it is AMG or REPA-style training. In their rebuttal, the authors attribute this to AMG stating "self-distilled targets act as a regularizer—adjusting the predicted flow via visual cues during training to rectify, rather than ignore, poor samples". I do not find this answer convincing since if it is true, the model should benefit from more (potentially noisy) data but Table 3 suggests otherwise and there is not experiments to back this claim.
>
> To summarize, my concern about the technical contribution of the paper still remains since the paper employs existing techniques with no added components or significant insights specific to addressed problem (V2A).
>
> **Qualitative results**
>
> I thank the authors for their clarifications. The authors indeed provided extensive quantitative results and a user study in the rebuttal to show the superiority of their method. However, the disagreement between the strong quantitative results and the poor qualitative results provided in the supplementaries makes me puzzled. I still find the provided qualitative results to be significantly lacking behind current SOTA methods and do not show the sufficient improvements with the provided baselines, especially with respect to the temporal alignment. I expect from a V2A paper to pay more attention with their insights are results to the temporal alignment which is the main challenge in V2A.
>
>
> **Sampling parameters**
>
> I thank the authors for providing the sampling parameters. My concern is partially resolved. Since the authors evaluate the models themselves, I believe it is more proper to evaluate at the same CFG or report best numbers for each baseline across CFG values. Multiple work showed that at high CFG values the FD increase rapidly and becomes misleading (e.g [1]).
>
>
> **Additional Clarification**
>
> Is the model trained separately from AMG and feature alignment in two stages or is there a typo in Eq. (7). shouldn't be FM-align?
>
> Overall, I find the paper to provide extensive experiments showing that the V2A benefits from existing diffusion techniques but no novel contributions or enough insights with regard to V2A, limiting the overall contributions of the paper. Based on the authors rebuttal response, I decided to raise my score.
>
> **References**
>
> [1] Analysis of Classifier-Free Guidance Weight Schedulers

---

> ### Author Response · Authors · 2025-08-04
>
> We sincerely thank the reviewer for the additional feedback and for raising the score. We truly appreciate your thoughtful evaluation of our rebuttal and supplementary material.
>
> **Sampling Parameters.**
> We’re glad that our responses helped address some of your concerns. For all compared methods, we used publicly available checkpoints and their default sampling parameters (including CFG values), following standard practice to ensure fair comparison. As you suggested, in the final version, we will additionally report results across different CFG values, including the best-performing configuration per method, to enhance transparency and completeness. We also acknowledge the concern about FD sensitivity at high CFG, and we will include further discussion in the revision following the analysis in [1].
>
> **Training Objective Clarification.**
> To clarify: our model is trained in a **single stage**, jointly optimizing both model guidance (AMG) and representation alignment losses as described in **Eq. (7)**. There is no separate or alternating training. We acknowledge the ambiguity arising from inconsistent terminology—specifically, the loss is referred to as $L_{FM-align}$ in L192 and later as $L_{AMG-align}$ in L211. In the final version, we will standardize the terminology and explicitly state in L211 that Eq. (7) is our final training objective, renaming it to $L_{FM-align}$ for clarity.
>
> **Novelty and Low-data Regime.**
> We understand your ongoing concern regarding novelty. Our goal is to adapt model guidance and multimodal alignment—techniques primarily explored in image generation domains—for the unique challenges of V2A. While these techniques are not entirely new, our contribution lies in their **joint integration** through a shared multimodal encoder trained end-to-end for V2A, along with extensive ablations that validate their effectiveness, particularly in low-data regimes. We appreciate your suggestion to further disentangle the effects of AMG and REPA in low-data regimes, and we will include more targeted ablations and analysis in future versions to provide deeper insights.
>
> **Qualitative Results and Temporal Alignment.**
> We appreciate your emphasis on temporal alignment, which indeed remains a central challenge in V2A. While our current framework does not explicitly optimize for temporal consistency, our additional experiments demonstrate that incorporating *temporally-aware visual conditioning* (e.g., FRIEREN-style) within our proposed framework significantly improves DeSync scores. This suggests that our model can serve as a strong, extensible foundation for future work focused on temporal alignment. We will emphasize this aspect more clearly in the revision and include additional qualitative comparisons that better reflect the quantitative gains.
>
> Thank you again for your constructive comments. We value your feedback and believe it has helped us strengthen the clarity, positioning, and future direction of our work. **We are especially grateful that you decided to raise your score, which means a great deal to us.**

---

### Official Review · Reviewer_q3sa · 2025-07-02

**Clarity:** 3
**Significance:** 2
**Originality:** 2
**Rating:** 4
**Confidence:** 4

**Summary:**

MGAudio, a model-guidance approach for conditional audio generation, replacing classifier-free guidance (during training). Adapted for video-to-audio, its dual-role alignment sets a new state-of-the-art on VGGSound. The paper is well-written, and shows some promising results.

**Questions:**

1. The authors mention throughout the paper (even early in the abstract) that they propose a “transformer denoiser”, but in their equation they work with velocity prediction. Can the authors clarify?
2. Even though the authors mention in L63-74 their contribution explicitly, it's hard to understand what is new in their method, and what was introduced before. Can the authors clarify this?
3. This paper introduces a new auxiliary loss, with a hyperparameter lambda, i couldn't find an ablation on this, and also not the actual value that was used.
4. “We introduce MGAudio, a novel framework for video-to-audio generation that replaces the conventional classifier-free guidance (CFG) objective with a model-guidance (MG) objective, leading to more efficient training and improved generation quality.” I find this statement misleading, as the method does use CFG to *sample* to the best of my understanding. Can you clarify if I misunderstood?

**Ethical Concerns:**

["NO or VERY MINOR ethics concerns only"]

**Final Justification:**

I had some concerns, they were addressed during the rebuttal.

**Limitations:**

yes

**Quality:**

3

**Strengths And Weaknesses:**

Strengths:

* The paper provides a thorough and convincing experimental evaluation: MGAudio is compared against a strong and recent set of baselines, including Diff-Foley, FRIEREN, and MDSGen, on two standard datasets.
* The paper is well written and easy to follow.

Weaknesses:
* Synergy between AMG and CFG: The paper finds that while MGAudio is trained with the AMG objective, applying CFG at inference time further boosts performance. This is somewhat counter-intuitive finding to me. The authors acknowledge this but didnt dig deeper into why.
* Computational cost: if i understand correctly, the claim about a single forward pass are regarding inference, but the authors do claim the CFG boost performance. It left me confused.

Overall, the results look promising and the method is valid. However, I am not sure if this paper belongs to neurips as the methodology is very modality-specific.

---

> ### Author Rebuttal · Authors · 2025-07-30
>
> We thank the reviewer for recognizing the promise of our results and the validity of our proposed method. Below, we address the weaknesses raised and respond to the reviewer’s questions in detail.
>
> ---
>
> > 1. **Weakness 1:** Synergy between AMG and CFG: The paper finds that while MGAudio is trained with the AMG objective, applying CFG at inference time further boosts performance. This is somewhat counter-intuitive finding to me. The authors acknowledge this but didnt dig deeper into why.
> >
> >    **Question 4:** “We introduce MGAudio, a novel framework for video-to-audio generation that replaces the conventional classifier-free guidance (CFG) objective with a model-guidance (MG) objective, leading to more efficient training and improved generation quality.” I find this statement misleading, as the method does use CFG to sample to the best of my understanding. Can you clarify if I misunderstood?
>
> Using CFG at inference after training with AMG is not contradictory. In fact, the observed performance improvement suggests that AMG **alleviates the training-inference mismatch**, enabling the model to more effectively benefit from CFG guidance.
>
> To clarify, one of the core motivations for introducing AMG is at L44–45:
> “...training to simulate both conditional and unconditional objectives. While effective, this multi-task setup may dilute the model’s capacity and lead to mismatched sampling behavior at inference.”
> AMG was specifically designed to compensate for this issue. It shows two key benefits in video to audio generation: (1) Mitigating training-inference mismatch by incorporating a CFG-like guidance term into training, making the model more compatible with CFG during inference; (2) Improving generation quality even without CFG by explicitly encouraging perceptual realism during training.
>
> These two effects are substantiated by our results. As illustrated in Figure 5 and Appendix Figure 4, MGAudio consistently outperforms both SiT and FRIEREN across both CFG and non-CFG settings, underscoring the effectiveness of AMG alone and its compatibility with CFG during inference.
>
> We acknowledge that this motivation was not sufficiently emphasized throughout the paper, and we thank the reviewer for bringing this to our attention. We will revise the manuscript to make this clearer.
>
>
> ---
>
> > 2. **Weakness 2:** Computational cost: if i understand correctly, the claim about a single forward pass are regarding inference, but the authors do claim the CFG boost performance. It left me confused.
>
> We believe the reviewer’s confusion stems from Lines 100–102, where we state that "...Vision Model-Guidance [16], achieving state-of-the-art FID on ImageNet with only a single forward pass at each inference step, half the computational cost of CFG." This was intended to highlight the **efficiency of vision model guidance (VMG) during inference**, in contrast to Classifier-Free Guidance (CFG), which typically requires two passes.
>
> To clarify:
>
> - MG (AMG) provides strong performance gains using **a single forward pass** during inference.
>
> - **Our method also supports CFG**, and combining MG with CFG can **further improve quality**.
>
> In prior work, MG showed strong improvements only when CFG was not used. In contrast, in our V2A setting, AMG consistently improves performance both with and without CFG at inference, as shown in Appendix Table 1 and Figure 4, outperforming the SiT baseline under both configurations.
>
> We will revise Lines 100–102 to make this distinction clear:
> MG enables efficient single-pass inference and is complementary to CFG, offering improvements in both inference modes.
>
> ---
>
> > 3. **Questions 1:** The authors mention throughout the paper (even early in the abstract) that they propose a “transformer denoiser”, but in their equation they work with velocity prediction. Can the authors clarify?
>
> As noted in L139–140, the transformer in our method can predict either noise or vector fields, depending on the framework. Since we adopt the flow matching approach, it predicts velocity. We agree that the term “transformer denoiser” may be misleading and will revise it to “transformer model” and clarify its role accordingly.
>
> ---
>
> > 4. **Questions 2:** Even though the authors mention in L63-74 their contribution explicitly, it's hard to understand what is new in their method, and what was introduced before. Can the authors clarify this?
>
> We thank the reviewer for highlighting the need to better clarify the novelty of our method. Below, we detail the key contributions of MGAudio and how they differ from prior work:
>
> #### 1. **Dual-Role Alignment of Pretrained Multi-Modal Representations**
>
> Our primary contribution is the **explicit dual-role utilization** of both audio and video branches from a pretrained multimodal encoder (CAVP) to guide generation:
>
> - The **video branch** is used for model-guided training (AMG),
>
> - The **audio branch** provides latent-space supervision via representation alignment (REPA).
>
> While prior works like Diff-Foley [19] and MDSGen [12] use the CAVP video encoder for conditioning, they discard the audio branch. Our method is the **first to jointly leverage both branches with distinct and complementary roles**. This dual-role integration improves both generation fidelity and training stability (see Sec. 3.2).
>
> This design is **non-trivial**: training a strong multi-modal representation model is computationally expensive, and naively using it as a condition vector does not leverage its full potential. We show that learned representation specifically for the target generating modality and guiding modality can be used to guide generation through the combination of model guidance (MG) and representation alignment (REPA). This presents a **promising, general framework for future cross-modal generation tasks**: first, train a powerful multi-modal representation learner, then leverage it for both trajectory guidance and generative modal internal feature alignment.
>
> #### 2. **Robust Learning from Noisy or Limited Data**
>
> We also show that MGAudio enables training on **limited and noisy AV pairs** by **self-distilling a target flow direction** informed by visual cues. This effect of improved robust learning from noisy or limited data has not been shown in the original vision-model-guidance paper [16]. Unlike CFG, which passively drops conditions, MGAudio actively corrects poor samples by injecting a plausible flow direction—effectively acting as a regularizer. As reviewer yNJc noted, this may explain the model’s ability to learn stable representations even with weak or noisy inputs. We support this with an experiment using only 10% of the VGGSound training set:
>
> **Table 1:** Comparison of fully trained MGAudio (with and without AMG) and MMAudio on the same randomly selected 10% subset of VGGSound.
> | Method  | AMG Loss | FAD ↓ | FD ↓   | IS ↑  | KL ↓ |
> |---------|----------|-------|--------|-------|------|
> | MGAudio (10%) | ✓        | 0.82  | 8.13   | 10.68 | 2.67 |
> | MGAudio (10%) | ✗        | 32.84 | 123.14 | 1.09  | 4.34 |
> | MMAudio (10%) | -        | 2.81  | 14.28  | 10.37 | 2.87 |
>
> This demonstrates MGAudio's effectiveness in regularizing training under data scarcity and its potential for data filtering or active sample weighting strategies in future work.
>
> ---
>
> > 5. **Questions 3:** This paper introduces a new auxiliary loss, with a hyperparameter lambda, i couldn't find an ablation on this, and also not the actual value that was used.
>
> As noted in Line 193, we set λ = 0.5 by default to weight the auxiliary **representation alignment loss** relative to the main generative objective. This follows the default from prior work [23]. To validate this choice, we conducted an ablation study across λ ∈ {0.25, 0.5, 0.75, 1.0}, using the same architecture and training setup (300k steps, batch size 32).
>
> **Table 2:** Ablation on λ the weight of the audio representation alignment loss.
> | Metric | λ = 0.25 | λ = 0.5 (default) | λ = 0.75 | λ = 1.0   |
> | ------ | -------- | ----------------- | -------- | --------- |
> | FAD ↓  | 0.74     | **0.71**          | 0.78     | 0.81      |
> | FD ↓   | 13.48    | 12.07             | 11.69    | **11.64** |
> | IS ↑   | 7.75     | 8.61              | **8.62** | **8.62**  |
>
> We observe that λ = 0.5 achieves the best perceptual quality (FAD), while higher λ slightly improves FD and IS at the cost of realism. The results suggest that λ = 0.5 offers a balanced trade-off, and the model remains robust across a reasonable range. We will include this ablation in the appendix of the revised version for clarity.

---

> > ### Comment · Reviewer_q3sa · 2025-08-05
> >
> > My concerns were mostly addressed. I will raise my score.

---

> > > ### Author Response · Authors · 2025-08-05
> > >
> > > Thank you very much for your thoughtful feedback. We are glad to hear that your concerns have been mostly addressed. **We sincerely appreciate your willingness to raise your score and your engagement throughout the review process.**

---

### Note · Authors · 2025-08-12

**We greatly appreciate that all reviewers increased their scores and recommended our work for acceptance.** We sincerely thank them for their constructive feedback and active engagement throughout the rebuttal period. We are also grateful for the thoughtful, encouraging, and responsive discussions, which have helped us further clarify and strengthen the contributions of our work.

---

### Decision · Program_Chairs · 2025-09-17

**Decision:**

Accept (poster)

**Comment:**

(a) The paper proposed a setup for real-world / open-domain video to audio generation; base on flow setups that uses video conditioning and also self 'correction' mechanism that relies on av encoder driving feature matching and conditioning. The resulting representations seem to generate better audio; The paper is written well; albeit some missing details that were later clarified.

(b) Although not entirely novel in its technical content, the paper passes the threshold on proposing a audio generation setup that is more intuitive for real world scenarios with video conditioning. The resulting model does deliver perceptually better content; And the approach will be potential impact on large-scale real-time audio and v2a generation setups. The work also builds on recent innovation in model guidance; in a way justifying that approach broadly for generative models.

(c) Hyperparameter optimization and architecture exploration is limited; and there is no explicit technical novelty in the paper; instead its really built on many recent innovations.

(d, e) The reviewers raised 3 main concerns most of which are addressed -- seemingly incremental results across the datasets/sota, no human validation results; and limited overall technical novelty. Per my read, I do not agree that technical novelty is below par for publication -- the paper's strengths are not in designing new algorithms, instead its on integrating recent innovations on an existing research problem, and showing progress; with potential impact on AV generation research. The authors also put quite a strong rebuttal re: results and their gains in performance.